# Comparison of the H₂O, HDO and δD stratospheric climatologies between the MIPAS-ESA v8, MIPAS-IMK v5 and ACE-FTS V4.1/4.2 satellite data sets

Karen De Los Ríos[1,2], Paulina Ordoñez[3,2], Gabriele P. Stiller[4], Piera Raspollini[5], Marco Gai[5], Kaley A. Walker[6], Cristina Peña-Ortiz[3], and Luis Acosta[1,7]

[1]Instituto de Física, Universidad Nacional Autónoma de México, Mexico City, 04510, Mexico
[2]Instituto de Ciencias de la Atmósfera y Cambio Climático, Universidad Nacional, Mexico City, 04510, Mexico
[3]Departamento de Sistemas Físicos, Químicos y Naturales, Universidad Pablo de Olavide, Sevilla, Spain
[4]Karlsruhe Institute of Technology, Institute for Meteorology and Climate Research, Hermann-von-Helmholtz-Platz 1, 76344 Leopoldshafen, Germany
[5]Istituto di Fisica Applicata "Nello Carrara" (IFAC) del Consiglio Nazionale delle Ricerche (CNR), Florence, Italy
[6]University of Toronto, Department of Physics, 60 St. George Street, Toronto, Ontario M5S1A7, Canada
[7]Instituto de Estructura de la Materia, CSIC, Serrano 121 28006, Madrid, Spain

*Correspondence to*: P. Ordoñez (p.ordonez.perez@gmail.com)

**Abstract.** Variations in the isotopologic composition of water vapour are fundamental for understanding the relative importance of different mechanisms of water vapor transport from the tropical upper troposphere to the lower stratosphere. Previous comparisons obtained from observations of H₂O and HDO by satellite instruments showed discrepancies. In this work, newer versions of H₂O and HDO retrievals from Envisat/MIPAS and SCISAT/ACE-FTS are compared. Specifically, MIPAS-IMK V5, MIPAS-ESA V8, and ACE-FTS V4.1/4.2 for the common period from February 2004 to April 2012 are compared for the first time through a profile-to-profile approach and comparison based on climatological structures. The comparison is essential for the scientific community to assess the quality of new satellite data products, a necessary procedure to validate further scientific work. Stratospheric H₂O coincident profiles analysis reveal good agreement between 16 km and 30 km. For HDO and δD, lower biases are found in the MIPAS-ESA and ACE-FTS comparison, even if associated to a larger de-biased standard deviation. The meridional cross-sections of H₂O and HDO exhibit the expected distribution that has been established in previous studies. For δD the tropical depletion in ACE-FTS and MIPAS-ESA occurs on the top of the dynamical tropopause, but this minimum is found at higher altitudes in MIPAS-IMK dataset. The tape recorder signal is present in H₂O and HDO for the three databases with slight quantitative differences. The δD annual variation for ACE-FTS data and MIPAS-ESA data is very weak compared to the MIPAS-IMK dataset, which shows a coherent tape recorder signal clearly detectable up to at least 30 km. The observed differences in the climatological δD composites between databases could lead to different interpretations regarding the water vapor transport processes toward the stratosphere. Therefore, it is important to further improve the quality of Level 2 products.

## 1. INTRODUCTION

Water vapour (WV) is the most important non-anthropogenic greenhouse gas in Earth's atmosphere (Hegglin et al., 2014). Although WV concentration is much lower in the stratosphere than in the troposphere, it significantly affects the climate at the surface (Solomon et al., 2007). Stratospheric water vapour (SWV) affects atmospheric dynamics and thermodynamics by modulating the radiative forcing directly (e.g., Solomon et al., 2010; Riese et al., 2012) and indirectly through its effect on the stratospheric ozone chemistry (Vogel et al., 2011). Moreover, it has been shown that the cold point temperature in the tropics is expected to rise in the future which will lead to increasing SWV due to reduced freeze-drying in the tropical tropopause layer (TTL; Gettelman et al., 2009). This implies the existence of a SWV feedback (Dessler et al., 2013; Banerjee et al., 2020).

The humidity in the lower stratosphere has been increasing in the last decades. Scientists discovered it at the beginning of the century. However, the reason for this humidification was not understood (Rosenlof et al., 2001). Because of the number of atmospheric composition measurement instruments that have been implemented on satellites over the past several decades, studies related to the SWV transport process have been increasing (e.g., Mote et al., 1996; Steinwagner et al., 2007; Lossow et al., 2011; Randel et al., 2012; Scheepmaker et al., 2016; Schneider et al., 2020). Brewer-Dobson circulation (Brewer, 1949) transports $H_2O$-rich air through upwelling from the troposphere at low latitudes, accompanied by large horizontal motions to mid-stratospheric latitudes. WV is also produced in the middle atmosphere through methane oxidation and is destroyed through photodissociation and reactions with O(1D) (Wang et al., 2018). However, the observed variability in SWV concentrations cannot be fully explained by observed changes in these main drivers (Hegglin et al., 2014). Therefore, studies focused on the dynamical processes that determine SWV variability constitute an active contemporary area of research (Plaza et al., 2021).

One way to conduct studies of troposphere-stratosphere mass transport is through isotopologues related to these species that behave as phenomenological tracers (Kuang et al., 2003). The isotopologic composition of WV molecules in the stratosphere provides an observational constraint for determining the relative importance of the possible transport mechanisms (Payne et al., 2007). Among the isotopologues of WV, $HD^{16}O$ (hereafter HDO) is particularly useful due to its significant fractionation effect (Merlivat and Nief, 1967; Kuang et al., 2003). Therefore, HDO at the tropopause is a very useful tracer to diagnose the relative importance of slow ascent and convective ice-lofting for WV transport into the stratosphere (Moyer et al., 1996; Tuinenburg et al., 2015; Wang et al., 2019).

Satellite remote sounding of the Earth's limb is currently the only method of providing near-global time series of atmospheric profiles from the upper troposphere to the lower thermosphere (Sheese et al., 2017). However, each atmospheric measurement with this method has its sources of uncertainty and systematic biases, which must be examined. Limb earth probing instruments may exhibit other systematic differences from similar devices depending on the observed latitudinal region and/or the observed local time. Sometimes, even significant discrepancies between data retrieved from the same satellite can be found depending on the algorithm. In general, all the biases have to be characterised and comparison between measurements retrieved from the same measurements or different instruments can provide insights on the quality of the data.

There are different datasets of WV and its isotopologues in the stratospheric region, retrieved mainly from three instruments. One of them, the Odin satellite, carries a Sub-Millimetre Radiometer (SMR), observing stratospheric $H_2O$, $H_2^{18}O$, and HDO (Murtagh et al., 2002). For technical reasons (the maximum bandwidth of a single radiometer is only 0.8 GHz), $H_2O$ and HDO cannot be measured simultaneously (Wang et al., 2018). Therefore, in this study this dataset will not be considered. The instrument MIPAS (Michelson Interferometer for Passive Atmospheric Sounding; Fischer et al., 2008) aboard Envisat (Environmental Satellite) was launched in 2002 and ceased operation in 2012. This instrument makes highly reliable WV observations in the stratosphere (Payne et al., 2007; von Clarmann et al., 2009; Ceccherini et al., 2011; Kiefer et al., 2023). The instrument ACE-FTS (Atmospheric Chemistry Experiment - Fourier Transform Spectrometer; Bernath et al., 2005; Nassar et al., 2005), aboard the Canadian satellite SCISAT, was launched in 2003 and yields WV information in the stratosphere to the present day (Boone et al., 2020).

In the case of MIPAS, different retrieval methods have been developed. One of the data sets, named here MIPAS-IMK, was retrieved with the IMK/IAA processor, which was developed in collaboration between the "Institut für Meteorologie und Klimaforschung" (IMK) in Karlsruhe, Germany, and the "Instituto de Astrofísica de Andalucía" (IAA) in Granada, Spain ( Speidel et al., 2018). The other MIPAS dataset, named here MIPAS-ESA V8 products (Dinelli et al. 2021), was retrieved by using the Optimized Retrieval Model (ORM) algorithm (Raspollini et al., 2022 and references therein) on the full-mission reprocessing campaign performed on L1V8 (Kleinert et al., 2018). The ACE-FTS retrievals have evolved through several versions with the retrieval model being updated with optimized parameters (Boone et al., 2005, 2013, 2020).

WV observations have been collectively evaluated through a multitude of parameters, like biases, drifts or variability characteristics, correlations, and other statistical data by the WCRP/SPARC water vapor assessment II (WAVAS-II) activity (https://amt.copernicus.org/articles/special_issue10_830.html). The last evaluation of Lossow et al. (2019) used ACE-FTS v3.5 (2004-2014), MIPAS-ESA V6 version (2002-2012) and V7 version (2005-2012), MIPAS-IMK V5H_H2O_20 (2002-2004) and V5R_H2O_220/221 (2005-2012). In this work, we use newer versions of some $H_2O$ data sets, namely MIPAS-ESA v8 and ACE-FTS V4.1/4.2, whose improvements will be described in the next section. Regarding HDO, Lossow et al. (2011) compared V5H_HDO_20 (2002-2004) retrieved with the IMK/IAA processor, SMR/Odin version 2.1, and ACE-FTS version 2.2, and they found good general agreement. However, distinct observational discrepancies of the δD (see section 3) annual variation were visible between MIPAS-IMK (Steinwagner et al., 2010) and ACE-FTS (Randel et al., 2012) data. Högberg et al (2019) assessed the profile-to-profile comparisons of stratospheric δD using two MIPAS-IMK sets from the retrieval based on V5H_H2O/HDO_20 and ACE-FTS V2.2 and V3.5. The overlap period was very limited, from February 2004 to March 2004. During this short overlap period, the majority of ACE-FTS observations occurred in March at northern polar latitudes and most of the coincidences are concentrated near 70º N. Lossow et al. (2020) reassessed the discrepancies in the annual variation δD in the tropical lower stratosphere based on MIPAS-IMK and ACE-FTS data sets. Overall, the used data set covered the period from July 2002 to March 2004, which is referred as the full resolution period of MIPAS. However, a longer time series is needed to draw robust conclusions on the relative importance of different mechanisms transporting WV into the stratosphere. Therefore, we use a new HDO data version called MIPAS-IMK V5H_HDO_22 (2002-2004) and

VR5_HDO_222/223 (2005-2012) that were first published by Speidel et al. (2018). Concerning MIPAS-ESA HDO, this was released on 2022 and there are no published comparisons yet. We focus here on the overlap period between MIPAS and ACE-FTS which is from 2004 to 2012.

    We compare the three $H_2O$, HDO and $\delta D$ databases relying on two approaches. First, we present profile-to-profile comparisons and provide a general overview of the typical biases in the observational databases. The second approach is based
on climatological comparisons, including meridional cross sections and time series comparisons. Section 2 describes the individual data sets in detail. In section 3, the methodology is outlined. Section 4 presents the results, which will be summarised in section 5.

## 2. DATA SETS

    As mentioned in the introduction, with the only exception of MIPAS-IMK $H_2O$ data, which use MIPAS-IMK
V5H_H2O_20 (2002-2004) and V5R_H2O_220/221 (2005-2012) as in Lossow et al. (2019), we employ newer data sets than those used in the previous studies. Here, we employ MIPAS-IMK V5H_HDO_22 (2002-2004) and V5R_HDO_222/223 (2005-2012) for HDO (Speidel et al., 2018), the MIPAS ESA Level 2 V8 dataset (Dinelli et al., 2021) results from the full-mission reprocessing campaign performed on L1V8 products and ACE-FTS V4.1/v.2 (Boone et al., 2020) for both isotopologues.

**2.1. MIPAS**

    MIPAS was a cooled, high-resolution Fourier transform spectrometer aboard Envisat (Fischer et al., 2008). Envisat was launched on 1 March 2002 and made observations until 8 April 2012, when communication with the satellite was lost. Envisat orbited the Earth 14 times a day in a sun-synchronous polar orbit at about 790 km altitude inclined of 98.55° with respect to the plane of the Equator. The equator crossing times were 10:00 and 22:00 local time for the descending and
ascending nodes, respectively. MIPAS measured the thermal emission of the atmospheric limb, covering all latitudes. MIPAS operated at 100% of its duty cycle from July 2002 to March 2004, when, due to a significant anomaly affecting the Interferometer Drive Unit (IDU), its regular operations were interrupted to avoid the mechanical blockage of the instrument (Dinelli et al., 2021). After various tests with different spectral resolutions, the European Space Agency (ESA) recovered the instrument in January 2005 at a reduced spectral resolution but a finer vertical sampling. At the beginning of 2005, MIPAS
operated at only a 30% duty cycle, which progressively increased until December 2007, when it was successfully restored to 100% operations (Kleinert et al., 2007, 2018). MIPAS operated in several observation modes regarding the altitude range covered and the width of the tangent altitude grid. Of relevance here are only the NOM (~5 to 72 km), UTLS-1 (~5 to 49 km), and the Aircraft emission (~7 to 38 km) observation modes.

### 2.1.1. MIPAS-ESA

130       The MIPAS ESA Level 2 V8 dataset (Dinelli et al., 2021) results from the full-mission reprocessing campaign performed on L1V8 products using the Optimized Retrieval Model (ORM) processor version 8.22 (Raspollini et al., 2022) funded by the European Space Agency (ESA). As a general approach, the retrieval algorithm fits modelled spectra to measured infrared spectra in species-dependent micro-windows via least-squares global fitting. For iteration control, the Gauss-Newton approach modified with the Levenberg-Marquardt method is used to minimize the fit residuals. Within the retrieval of data

from the second phase of MIPAS operation, regulation is needed because the tangent altitude steps were smaller than the field-of-view width so that the spectra along a vertical profile were not independent, and the inversion problem was underdetermined for retrieval of a value at each tangent height. The regularization is applied a posteriori in case of $H_2O$ with a retrieval error-dependent regularisation strength (Ridolfi and Sgheri, 2011). HDO was retrieved for the first time within the V8 data set. The retrieval is set up as optimal estimation retrieval. The a priori used is the previously retrieved $H_2O$ profile, scaled by the

constant isotopic ratio used by the HITRAN spectroscopic database VSMOW (see Sect.3). The diagonal elements of the covariance matrix of the a priori which determine the strength of the regularization are computed as the square of the sum of a constant ($10^{-3}$ ppmv) plus the 100% of the a priori profile. This choice assures that the assumed uncertainty of the a priori is at least 100% of the a priori profile or 1 ppbv squared, whatever is larger, to keep the regularization strength low. The non-diagonal elements are computed assuming a correlation length of 10 km in the vertical. HDO has been retrieved from all the

observation modes listed above; the useful altitude range is reported to be 5 to 55 km (Raspollini et al., 2021). The microwindows used for the retrieval of HDO lie in the 1218 $cm^{-1}$ to 1471 $cm^{-1}$ spectral range, while the ones used for the retrieval of $H_2O$ lies in the ranges 783 to 956 $cm^{-1}$ and 1224 to 1696 $cm^{-1}$.

      MIPAS ESA L2 analysis uses the HITRAN_mipas_pf4.45 spectroscopic database. It is based on HITRAN08 (Rothman et al., 2009), but spectroscopic parameters for the molecules $H_2O$, $O_2$, $SO_2$, OCS, $CH_3Cl$, $C_2H_2$ and $C_2H_6$ are taken

from HITRAN 2012 (Rothman et al., 2012). The spectroscopic parameters of $HNO_3$ were derived by Perrin et al. (2016), and the spectroscopic data for $COCl_2$ were derived by Tchana et al. (2015). Both $HNO_3$ and $COCl_2$ data are now contained in HITRAN 2016 (Gordon et al., 2017).

      The estimation of the systematic error of MIPAS-ESA $H_2O$ and HDO profiled can be found at http://eodg.atm.ox.ac.uk/MIPAS/err/err_hdo_day_or27.png and http://eodg.atm.ox.ac.uk/MIPAS/err/err_h2o_day_or27.png

respectively. Noise error, Averaging Kernels, vertical resolution are discussed at https://earth.esa.int/eogateway/documents/20142/37627/README_V8_issue_1.1_20210916.pdf.

      $H_2O$ vertical resolution is about 3 at 10 km, then it slowly degrades, reaching 5 - 6 km at 20 km, 7.5 at 30 - 40 km, 10 at 50 km. The total random error is about 1-2% in the range 50 hPa-1 hPa for all atmospheres except polar winter, where it may reach values even larger than 5%. The tropopause is characterized by large percent random noise (also due to the minimum

of the VMR), in the mesosphere random error rapidly increases with the altitude. HDO vertical resolution is 3-3.5 km in the

range 6-10 km, about 5 km in the range 6-30 km, it is 7.5 km at 40 km and 12.5 km at 50 km. The relative average single scan random error varies with altitude for the different atmospheres, but it is never smaller than 25%.

### 2.1.2. MIPAS-IMK

The MIPAS-IMK database is a product of the collaboration between IMK and IAA who developed an algorithm for
the retrieval of the VMR of about 30 different trace gases from MIPAS level-1b data independent of the ESA algorithm (von Clarmann et al., 2009). Similar to the MIPAS-ESA product, the IMK-IAA algorithm uses a non-linear least-squares global-fitting technique with Levenberg-Marquardt damping to fit simulated spectra to measured ones within spectral microwindows where the respective species have suitable spectral lines. In contrast to the MIPAS-ESA approach whose retrieval grid coincides with the tangent altitudes of the measurements, the level-2 data are retrieved on a fixed grid of 1 km step up to 46
km and 2 km above. This grid width again requires regularization to stabilize the retrieval. A Tikhonov regularization was chosen that acts as a smoothing constraint by weighted minimization of the squared first order finite differences of adjacent profile values. The regularization strength was chosen to optimize vertical resolution while limiting unphysical oscillations in the retrieved profiles. The MIPAS-IMK WV retrievals used here were retrieved in log (VMR) space from V5 MIPAS spectra (see e.g., the SPARC-WAVAS-II Special issue (https://amt.copernicus.org/articles/special_issue10_830.html) for validation
of this data version, V5H_H2O_20 and V5R_H2O_220/221). The HDO data version used in this study differs significantly from the data versions assessed by Lossow et al. (2020) and Högberg et al. (2019) and used by Steinwagner et al. (2007, 2010). For the data version used here (V5H_HDO_22 and V5R_HDO_222/223), HDO was retrieved in linear space with the previously retrieved main isotopologue profile, scaled by the constant isotopic ratio used in the HITRAN database (VSMOW) as a priori information. $\delta$D (see section 3) is calculated from the regular water vapour product and HDO; by this new approach
the disadvantage of using a less-than-optimal data version of $H_2O$ is omitted, and the vertical resolution of $\delta$D is provided by the difference between the a priori and the retrieved profile (Speidel et al., 2018). By this change of the retrieval approach the disadvantages of the previous HDO and $\delta$D data product demonstrated by Lossow et al. (2020) should be overcome. MIPAS-IMK V5H_HDO_22 (2002-2004) and V5R_HDO_222/223 (2005-2012) data are available from NOM observation mode only, leading to a lower number of total available profiles than for ESA data. Spectral microwindows in the 1250 to 1482 cm$^{-1}$ range
were used for the HDO retrieval while $H_2O$ was retrieved in the 795 to 827 cm$^{-1}$ and 1224 to 1410 cm$^{-1}$ spectral range. Spectroscopic data from the MIPAS-specific data base MIPAS_pf3.32 were used, which are, for $H_2O$ and HDO, essentially the same data as in its earlier version published by Flaud et al. (2003) and based in general on the HITRAN1996 data base (Rothman et al., 1998). Differences for $H_2O$ and HDO between MIPAS pf3.32 and HITRAN1996 are updates that were available at the HITRAN web site at the time when the MIPAS-specific data base was collected, and parameters for the main
isotopologue derived from recent theoretical calculations (for more details, see Flaud et al. (2003)).

Information on systematic errors, averaging kernels and vertical resolution of $H_2O$ can be found in von Clarmann et al., 2009. In summary, the vertical resolution is between 2.3 km in the lower and 6.9 km in the upper stratosphere. The

systematic errors are dominated by spectroscopic uncertainties and are in the order of 7 to 19%. For HDO, the estimated random errors are between 15% at about 15 km and 35% at 40 km altitude, and the vertical resolution increases from 3 to 4 km up to 25km to 6 km at 35 km. The averaging kernels are well behaved, i.e. peak at the nominal retrieval height, between 15 and 40 km. The systematic errors are again dominated by spectroscopic uncertainties.

## 2.2. ACE-FTS

ACE-FTS is one of three instruments aboard the Canadian satellite SCISAT (Bernath et al., 2005). SCISAT was launched on the 12 August 2003 into a highly inclined, 74° orbit at 650 km altitude. This orbit provides latitudinal coverage of 85° S to 85° N but is optimized for observations at high and middle latitudes. ACE-FTS measures the Earth's atmosphere during up to 15 sunrises and 15 sunsets daily, from approximately 5 to 150 km altitude. Vertical sampling varies with altitude and orbit beta angle, from a minimum of around 1 to 2 km in the upper troposphere up to a maximum of approximately 6 km in the upper stratosphere and mesosphere. HDO information is retrieved from two spectral bands: 3.7 to 4.0 µm (2493-2673 cm$^{-1}$) and 6.6 to 7.2 µm (1383-1511 cm$^{-1}$). $H_2O$ retrieval uses spectral information between 3.2 and 10.7 µm (937-3173 cm$^{-1}$) (Boone et al., 2005).

Here, we use ACE-FTS version 4.1/4.2. The ACE-FTS trace species VMR retrieval algorithm is described by Boone et al. (2005, 2013), and the changes for the version 4.1/4.2 retrieval are provided in Boone et al. (2020). Similar to MIPAS, the retrieval algorithm uses a non-linear least-squares global-fitting technique that fits forward modelled spectra to the ACE-FTS observed spectra in given microwindows - based on line strengths and line widths from the HITRAN 2016 database (with updates as described by Gordon et al. (2017)). The pressure and temperature profiles used in the forward model are the ACE-FTS derived profiles, calculated by fitting $CO_2$ lines in the observed spectra. The version 4.1/4.2 retrieval grid uses a minimum altitude spacing of 2 km for tangent heights above 15 km and a minimum spacing of 1 km for tangent heights below 15 km. This limitation on the retrieval grid suppresses unphysical oscillations that commonly occurred above 15 km in previous processing versions when the tangent height spacing dropped below 2 km. The main changes made in the v4 retrievals are updated micro windows for most species that allow for a more significant number of interfering species; improvements to the temperature and pressure retrievals, leading to fewer unnatural oscillations in the vertical profiles (Sheese et al., 2017).

## 3. METHODS

All data used here were managed in agreement with the user manuals of each dataset. For MIPAS-IMK, we followed Lossow et al. (2020) and Högberg et al. (2019). For ACE-FTS, we used the specifications given by Sheese et al. (2015) and Boone et al. (2020). Dinelli et al. (2021) was employed for MIPAS-ESA. The present quality assessment of $H_2O$, HDO and $\delta D$ data mainly focuses on the stratosphere, although data for the upper troposphere and lower mesosphere are used if available.

For calculating δD, we assessed the isotopic composition through the expression $R = \frac{[D]}{[H]}$ that can be determined through the concentration of the isotopologues of water as follows:

$$R = \frac{[D]}{[H]} = \frac{[HDO] + 2[DDO]}{2[H_2O] + [HDO]} \approx \frac{[HDO]}{2[H_2O]} \qquad (1)$$

To quantify the abundances of heavy isotopes, $R$ is usually compared to a standard reference ratio known as $R_{VSMOW}$ through the following relationship:

$$\delta D = \left(\frac{R}{R_{VSMOW}} - 1\right) \times 1000 \qquad (2)$$

where VSMOW=155.76 x $10^{-6}$ is the reference ratio (Vienna Standard Mean Ocean Water; Hagemann et al., 1970).

### 3.1. Profile to profile comparisons

This approach is based on the comparison of averages of coincident profiles. For the coincidence pairs, the ACE-FTS data, which is the sparser dataset in the tropics was used as the first data set. Then, ACE-FTS and MIPAS-ESA observations are considered to be coincident when they meet the following criteria (Högberg et al., 2019):

- Spatial separation of less than 1000 km.
- Temporal separation less than 24 h.

- Geolocation separation less than 5°, both in longitude and equivalent latitude.

The MIPAS-IMK profile coincident with the selected MIPAS-ESA profile was found by using the following criteria:

- Temporal separation less or equal to 2 seconds.
- Geolocation separation less than 1° in latitude.

Figure 1 shows a map for all the coincident profiles between 25-27 July 2010, illustrating that for each ACE-FTS profile (green diamond), there is one MIPAS-IMK (red square) and one MIPAS-ESA (blue asterisk) profile that meet the coincidence criteria. Only data points with the full triple of observations (ACE-FTS, MIPAS-ESA, and MIPAS-IMK) were used for direct comparisons described below.

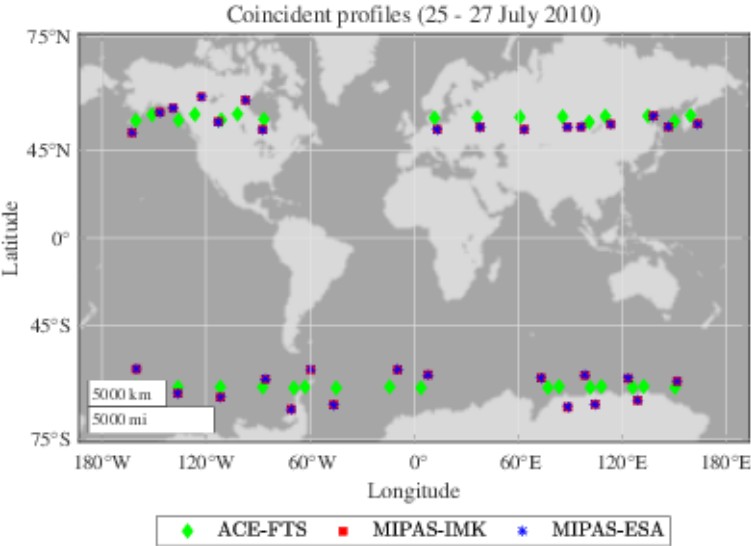

**Figure 1.** Coincident profiles between ACE-FTS, MIPAS-ESA and MIPAS-IMK on 25-27 July 2010. Different markers indicate the
database, ACE-FTS (green diamonds), MIPAS-IMK (red squares), and MIPAS-ESA (blue asterisks).

The profiles from each dataset were linearly interpolated for the comparisons onto a 58 levels grid from 1 to 70 km
(1 km grid from 0 to 44 km and 2 km step width from 46 to 70 km), which are the altitude reference levels of MIPAS-IMK as
described by Lossow et al. (2011). Fig. 2 shows the number of matched profiles by altitude between ACE-FTS, MIPAS-ESA
and MIPAS-IMK. The number of valid matches increases in the UTLS, and more than 10,000 matched profiles are obtained
from the mid-stratosphere. The number of ACE-FTS HDO profiles decreases from 40 km of altitude and upwards. At 48 km
of altitude the last profile is found.

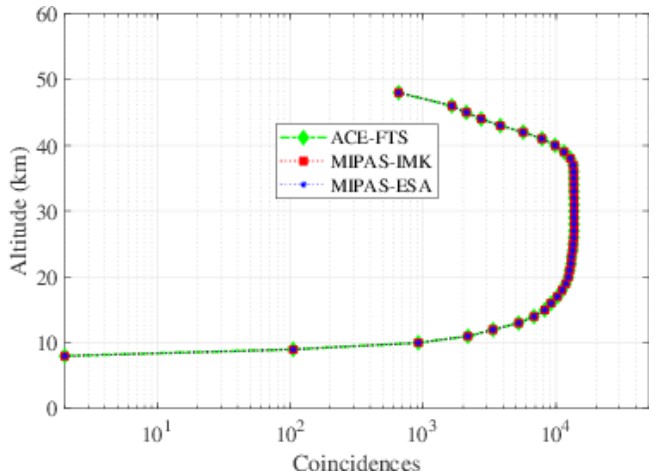

**Figure 2.** The number of coincident sets of data (2004-2012) for ACE-FTS, MIPAS-ESA and MIPAS-IMK comparisons.

Then, the mean is computed as the arithmetical average of the data distribution for each altitude level and the data dispersion is obtained by the standard error of the mean (SEM), i.e, the standard error divided by the square root of the sample. $\delta D$ can be quantified by two approaches: (1) calculate R from individual HDO and $H_2O$ profiles and average the results, or (2) first compute the averages values of $H_2O$ and HDO from all the profiles and then calculate R. In this work, the second approach is used as defined by Högberg et al. (2019) and Lossow et al (2020).

### 3.1.1. Bias determination

Four statistical parameters have been calculated globally at each altitude level: mean absolute biases, mean relative biases, the de-biased standard deviation of the differences and Pearson correlation coefficient. The mean bias between two coincident data sets for a specific altitude level has been calculated as:

$$\bar{b}(z) = \frac{1}{n(z)} \sum_{i=1}^{n(z)} \delta_i(z) \tag{3}$$

where $n$ denotes the corresponding number of coincident measurements, and $z$ the altitude. These differences are considered as:

$$\delta_i(z) = \frac{x_i(z)_1 - x_i(z)_2}{x_i(z)_{ref}} \tag{4}$$

where $x_i(z)_1$ are the individual $H_2O$, HDO or $\delta D$ abundances of the first data set and $x_i(z)_2$ are the abundances of the second data set that are compared.

**The mean absolute bias:**

This is calculated when $x_i(z)_{ref} = 1$ for absolute analysis in Eq. (4)

$$\overline{b_{abs}}(z) = \frac{1}{n(z)} \sum_{i=1}^{n(z)} x_i(z)_1 - x_i(z)_2 \tag{5}$$

**The mean relative bias:**

This is calculated by dividing the mean absolute bias by the mean reference value (Wetzel et al., 2013). For the reference value, different options are possible (e.g., Randall et al., 2003; Dupuy et al., 2009). The mean of the two datasets have been chosen because the satellite observations can have large uncertainties, and thus the mean is an appropriate approach (Lossow et al., 2019):

$$x_{i_{ref}}(z) = \frac{x_i(z)_1 + x_i(z)_2}{2} \tag{6}$$

Then, considering the Eq. (3) and (4) the mean relative bias is given by:

$$\overline{b_{rel}}(z) = \frac{1}{n(z)} \sum_{i=1}^{n(z)} 2\left(\frac{x_i(z)_1 - x_i(z)_2}{x_i(z)_1 + x_i(z)_2}\right) \tag{7}$$

**De-biased standard deviation:**

The de-biased standard deviation ($\sigma_{\overline{b}}$) is represented by the standard deviation of the mean relative bias-corrected between the two sets of compared data:

$$\sigma_{\overline{b}}(z) = \sqrt{\frac{1}{n(z) - 1} \sum_{i=1}^{n} \left(\delta_i(z) - \overline{b(z)}\right)^2} \tag{8}$$

This quantity measures the precision of the relative bias between the two datasets being compared, particularly in cases where a complete evaluation of the random error budget is not available for all the instruments involved (von Clarmann et al., 2006).

**Pearson correlation coefficient:**

The correlation coefficient $r$ dependent on altitude levels is defined as:

$$r(z) = \frac{1}{n(z) - 1} \sum_{i=1}^{n(z)} \left(\frac{x_i(z)_1 - \overline{x(z)_1}}{\sigma_{x_1}}\right)\left(\frac{x_i(z)_2 - \overline{x(z)_2}}{\sigma_{x_2}}\right) \tag{9}$$

Where $\sigma_{x_1}$ and $\sigma_{x_2}$ are the standard deviation of the first and the second dataset abundances respectively. We use this standard methodology because the quantity of data is large in all cases, and then the data distribution behaves as a normal distribution, resulting in a robust correlation coefficient (Lanzante, 1996).

### 3.2. Other comparisons as a function of space and time

Here we compare the climatologies of $H_2O$, HDO and $\delta D$. In this approach, each grid box represents an average over several measurements. It has the advantage of not requiring coincidences. This approach has the bonus that the used data sets are larger, but a weakness is that sampling biases can affect the comparison.

We first performed the data binning. $x_i$ ($\overline{\theta}, t, z$) is the individual concentration of $H_2O$, HDO or $\delta D$ for a given time $t$, a latitude $\overline{\theta}$ and for an altitude $z$. We average the data sets that match the condition for belonging to a given bin.

$$I_{\overline{VMR}}(\overline{\theta}, t, z) = \frac{1}{n_o} \sum_{i=1}^{n_o} x_i(\overline{\theta}, t, z) \tag{10}$$

where $n_o$ is the amount of data found within the established grid, and $I_{\overline{VMR}}$ is the value representing all the data fulfilling the grid condition (Högberg et al., 2019).

From the grid box means of $H_2O$, HDO and δD, some climatologies are compared for the period 2004-2012. The first one is a comparison of latitude - altitude cross sections (zonal means) for a time interval. We analyzed zonal means constructed from 10º latitude bins over the seasons December to February and June to August. Examining time series is another way to compare the data. The time series used in this section are based on monthly zonal means obtained considering the latitude range from 30º S to 30º N for each month. This comparison shows how each database captures seasonal and annual cycles.

## 4. RESULTS AND DISCUSSION

### 4.1. Vertical profiles comparisons

The $H_2O$ average profiles for ACE-FTS, MIPAS-IMK and MIPAS-ESA computed on all coincident profiles (2004-2012) and all latitudes are shown in Fig. 3 (a). The bars given for the average profiles are the standard error of the mean (SEM) distribution of measurements at each altitude level. The profiles of $H_2O$ exhibit a slight increase with altitude in the stratosphere (from 15 km up to 50 km, approximately) both for MIPAS-IMK and MIPAS-ESA, which is consistent with the stronger chemical generation of WV through methane oxidation in the upper stratosphere near 50 km (LeTexier et al., 1988). ACE-FTS average profiles are consistent with the two MIPAS profiles up to 30 km, in particular ACE-FTS and MIPAS-IMK profiles are almost identical between 20 and 30 km (Fig. 3 (a)). However, above 30 km, ACE-FTS $H_2O$ profiles have a significant deviation from the other two databases, which was also found in the SPARC/WAVAS-II comparisons for earlier data versions with respect to many other satellite data records (see e.g., Lossow et al., 2019). All $H_2O$ average profiles have a minimum around the tropopause at 17 km of altitude.

The average HDO vertical profiles, along with the standard error of the means are shown for the three databases in Fig. 3 (b). ACE-FTS and MIPAS-IMK average profiles are almost identical in the range between 12 and 48 km. MIPAS-ESA dataset is almost identical to the other two datasets in the lower stratosphere (13 to 34 km) but exhibits a dry bias in the upper stratosphere (i.e., above 34 km, see Fig. 3 (b)). ACE-FTS and MIPAS-IMK have a minimum concentration at 16 km of altitude, while the coincident profiles of MIPAS-ESA have a minimum around 12 km. Högberg et al. (2019) also compared HDO profiles to previous versions of the MIPAS-IMK and ACE-FTS for the period February–March 2004. They demonstrated a high consistency in the structures along the stratosphere between the two databases. They also showed a dry bias of MIPAS-IMK in the tropopause region, which does not exist in this new version of the data.

The average δD vertical profiles of the three databases are in reasonable agreement from 13 to 30 km of altitude (Fig 3 (c)). Above 30 km ACE-FTS δD mean profile shows a positive bias compared to the two MIPAS databases, probably derived from the dry bias of ACE-FTS $H_2O$ data. On the other hand, MIPAS-ESA δD depicts a negative bias from 33 km upwards, probably derived from the MIPAS-ESA HDO dry bias at these altitudes. The optimal level of agreement between the three data sets is observed in the altitude range between 16 and 30 km, to which we will restrict the climatological comparisons.

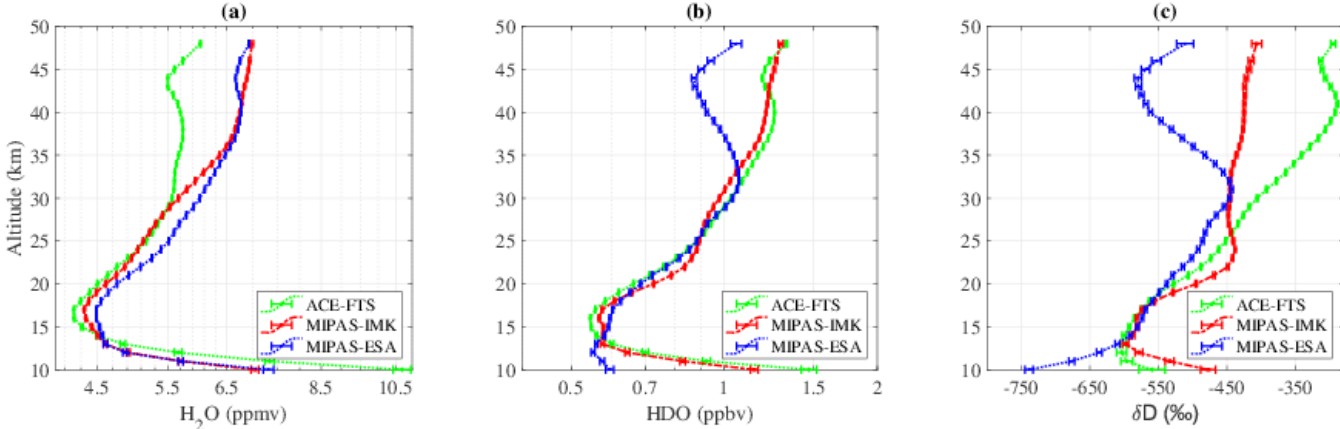

**Figure 3.** Averaged vertical profiles comparison between ACE-FTS (green diamonds), MIPAS-IMK (red squares) and MIPAS-ESA (blue asterisks) for (a) $H_2O$ observations, (b) HDO observations and (c) $\delta D$. The bars represent the standard error of the mean.

## 4.2. Bias comparison

Figure 4 shows the biases derived from the profile-to-profile comparisons described in section 3.1.1. As shown above, the comparisons are typically based on several thousand coincidences above approximately 15 km, and cover latitudes from 90 S to 90 N for the 2004-2012 period.

For $H_2O$, there is a good agreement between the three datasets in the altitude range between 16 km and 30 km (Fig. 4 (a) for absolute differences and Fig. 4 (b) for percent differences), with maximum percent differences of 8.2% between ACE-FTS and MIPAS-ESA, and maximum percent differences smaller than 3.7% between MIPAS-IMK and ACE-FTS. Differences between MIPAS-ESA and MIPAS-IMK could be ascribed at least in part to the different spectroscopic databases used by the two algorithms, since they resemble the differences in $H_2O$ profile retrieved with spectroscopic database sp4.45 (used by MIPAS-ESA) and sp3.2 (used by MIPAS-IMK) (Ridolfi, private communications). Above 30 km, the bias between MIPAS (both ESA and IMK) and ACE-FTS increases with altitude reaching values exceeding 20% around approximately 40 km, where the bias starts to decrease. Figure 4 (c) shows the de-biased standard deviations of $H_2O$ obtained by comparing the datasets. All the data sets show a good agreement and small variations in terms of spread in the whole stratosphere. It is worth mentioning that lower de-biased standard deviations are found in the MIPAS-ESA to ACE-FTS comparison above 25 km of altitude coupled with the higher correlations of these datasets (Fig 4 (d)), which indicates that the MIPAS-IMK retrievals are less sensitive to actual atmospheric variations in $H_2O$ than the other two data sets above 25 km.


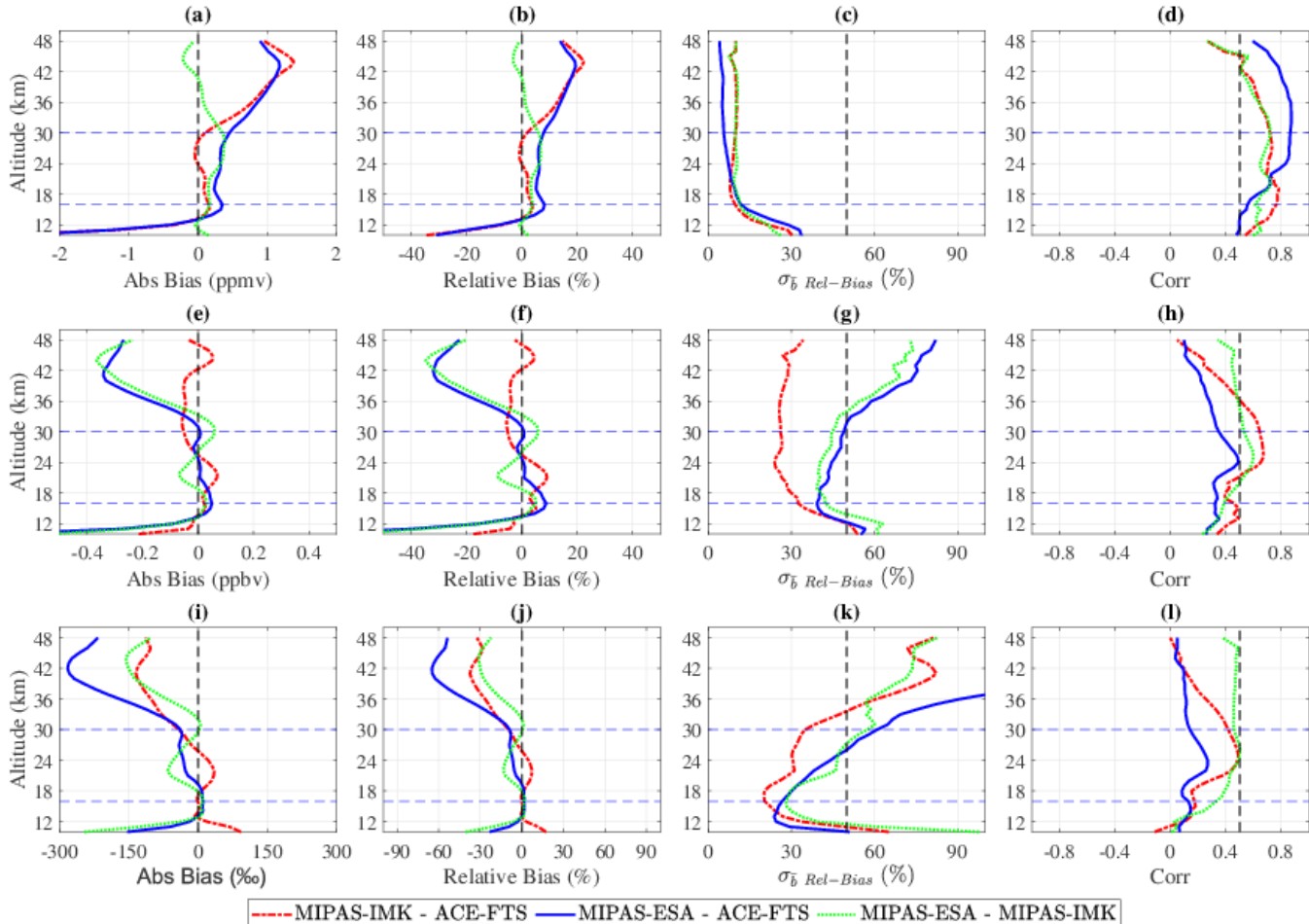

**Figure 4.** Comparisons between MIPAS-IMK, MIPAS-ESA and ACE-FTS for $H_2O$ (top panels), HDO (middle panels) and δD (bottom panels): (a, e, i) the absolute bias, (b, f, j) the relative bias, (c, g, k) the de-biased standard deviation of the relative bias and (d, h, l) correlation coefficients. Black dashed lines indicate 0 ppbv, 0%, 50% and 0.5 from left to right in the different panels. Blue dashed lines show the 16 km and the 30 km levels. Maximum and minimum values obtained for the range 16−30 km are indicated in Table 1.

HDO absolute differences between the three data sets are within ± 0.1 ppbv in the 16 to 30 km altitude range (Fig. 4 (e)), corresponding to 9.1% in relative terms (Fig. 4 (f)). Above 30 km MIPAS-ESA shows a dry bias with respect to both MIPAS-IMK and ACE-FTS which could be related to spectroscopic errors, providing the largest contribution to MIPAS-ESA HDO systematic errors in the altitude range above 30 km. The HDO de-biased standard deviations (Fig. 4 (g)) show values lower than 49%. Conversely to $H_2O$, the lower de-biased standard deviations for HDO are found for the MIPAS-IMK to ACE-FTS comparison above the 15 km region, which is coupled with its higher correlation coefficients and indicates that the MIPAS-ESA retrieval are either less sensitive to atmospheric variations of HDO or uses a weaker regularisation. The larger

spread of the differences when MIPAS-ESA data set is involved is consistent with the MIPAS-ESA random error larger than MIPAS-IMK.

       The comparison of $\delta$D (see Fig. 4 (i) for absolute differences and Fig. 4 (j) for percent differences) shows an agreement within 8.5 % between ACE-FTS and MIPAS-ESA and within 13.4% for MIPAS-ESA and MIPAS-IMK in the range between 16 km and 30 km approximately. Larger biases are found above 30 km where the largest deviations are found in the MIPAS-

ESA and ACE-FTS comparisons, due to ACE-FTS negative bias in $H_2O$ and MIPAS-ESA negative bias in HDO. The smaller relative de-biased standard deviation in the lower and the middle stratosphere (Fig. 4 (k)) is found for ACE-FTS and MIPAS-IMK comparison (between 20 and 34%), consistent with the larger random noise of MIPAS-ESA HDO. Pearson correlation coefficients are greater than 0.4 with the comparisons between MIPAS-ESA and MIPAS-IMK datasets (Fig. 4 (l)). The correlation coefficients in the $\delta$D comparisons of the ACE-FTS and MIPAS-ESA data show the lowest agreement with values

in the range of 0.1 and 0.2 for the lower and middle stratosphere.

       These results are in accordance with comparisons by Högberg et al. (2019) between MIPAS-IMK and ACE-FTS which were performed with previous versions of the data for a very limited overlap period (from February 2004 to March 2004), where relative biases for $H_2O$, HDO and $\delta$D were found to be smaller than 10 % in the middle stratosphere. However, in our current study, $\delta$D deviations in the UTLS region show lower values than the biases founded by these authors. For

MIPAS-ESA, Raspollini et al. (2020) also showed the HDO mean absolute and relative bias between MIPAS-ESA and ACE-FTS V4.1/4.2 data for each year from 2004 to 2012. Even if different coincidence criteria are used for the determination of coincident profiles, their results are in agreement with our results suggesting a dry bias of MIPAS-ESA HDO above 30 km, an agreement within 10% in the altitude range 16-30 km, and a dry bias below 12 km.

       In order to understand the differences between the two MIPAS databases and the fact that, in some cases, the two

MIPAS datasets are more different than MIPAS and ACE-FTS we have to consider that there are differences in the algorithms, in the selected spectral points, but also in the used spectroscopic database (MIPAS-ESA using spectroscopic data for $H_2O$ and HDO based on HITRAN 2012, while MIPAS-IMK using data based on HITRAN 2008) and in the used radiances (MIPAS-ESA using the last release of L1V8 data, while MIPAS-IMK using L1V5 data). L1V8 data have been corrected with an upgraded radiometric calibration (Kleinert et al., 2018), impacting both the radiance and its temporal drift.

395        Discrepancies in the troposphere and upper levels of the stratosphere derived from the bias analysis indicate that the three databases are in good agreement only between 16 to 30 km. Therefore, in the following, the climatological analysis will be restricted to the range of the lower and the middle stratosphere. Table 1 summarises the dataset average characteristics of the $H_2O$, HDO and $\delta$D comparisons between 16 to 30 km for the period 2004-2012. The results come from coincident profiles for the full globe without latitude restriction.


**Table 1**. $H_2O$, HDO and δD range of the statistical quantities for the comparison of the databases between 16 to 30 km of altitude for the full globe as summary of the Fig. 4. Absolute bias (Abs. bias), relative bias (Rel. bias), De-biased standard deviation (De-biased SD) and Pearson correlation coefficient (r) values are indicated.

| | | MIPAS-IMK – ACE-FTS | MIPAS-ESA – ACE-FTS | MIPAS-ESA – MIPAS-IMK |
|---|---|---|---|---|
| **Abs. Bias** | **$H_2O$ (ppmv)** | −0.05 to 0.16 | 0.23 to 0.45 | 0.15 to 0.38 |
| | **HDO (ppbv)** | −0.05 to 0.07 | −0.02 to 0.05 | −0.07 to 0.06 |
| | **δD (‰)** | −40.75 to 34.96 | −41.22 to 10.45 | −65.59 to 11.28 |
| **Rel. Bias (%)** | **$H_2O$** | −0.9 to 3.7 | 5.1 to 8.2 | 3.2 to 6.8 |
| | **HDO** | −5.1 to 9.1 | −1.8 to 8.7 | −8.7 to 5.9 |
| | **δD** | −9.4 to 7.3 | −8.5 to 1.8 | −13.4 to 2.0 |
| **De-biased SD (%)** | **$H_2O$** | 7.9 to 9.9 | 5.6 to 11.8 | 9.2 to 11.4 |
| | **HDO** | 24.0 to 32.5 | 39.3 to 49.0 | 39.2 to 44.6 |
| | **δD** | 20.1 to 34.0 | 26.7 to 60.9 | 28.0 to 57.4 |
| **r** | **$H_2O$** | 0.7 to 0.8 | 0.6 to 0.9 | 0.6 to 0.7 |
| | **HDO** | 0.4 to 0.7 | 0.3 to 0.5 | 0.4 to 0.6 |
| | **δD** | 0.2 to 0.5 | 0.1 to 0.3 | 0.3 to 0.5 |


$H_2O$ biases in the lower and middle stratosphere (16 km to 30 km) ranged from -0.05ppmv to 0.45 ppmv across the three databases. Lower $H_2O$ absolute and relative biases are found for the MIPAS-IMK–ACE-FTS comparison with values up to 0.16 ppmv and 3.7% respectively. Lower de-biased standard deviations (up to 9.9%) and higher correlation coefficients (from 0.7 to 08) are found in the same databases comparison. Therefore, $H_2O$ profiles from MIPAS-IMK and ACE-FTS are

in better agreement than compared to MIPAS-ESA dataset. For HDO, lower absolute and relative biases are found in the MIPAS-ESA–ACE-FTS comparisons with values up to 0.05 ppbv and 8.7% respectively. The lower δD values for both absolute and relative biases are found for the same comparison as HDO (MIPAS-ESA–ACE-FTS) up to 41.2‰ and 8.5% respectively. However, the de-biased standard deviations reach values of 49% and 61% for HDO and δD respectively. Only the comparison between the correlation coefficients for δD obtains the better result for the two MIPAS datasets with values

from 0.3 to 0.5.

## 4.3. Comparisons of seasonally averaged latitude cross-sections

Figure 5 shows the seasonally averaged latitude-altitude cross sections of $H_2O$, HDO and δD for the three data sets from 80 S to 80 N. The maps are performed from all data available without coincidence considered. Water vapour shows a large depletion in the tropopause in the three datasets both in JJA (Fig. 5(a)) and DJF (Fig. 5 (b)), with values between 3 and

5 ppmv in the lower stratosphere. The depletion in the tropics occurs at a higher altitude than in the mid-latitudes. A secondary minimum in the tropical middle stratosphere is also appreciated in both seasons, associated with the minimum originating in the lower stratosphere during the previous year and propagated upward by the Brewer-Dobson circulation. This ascent of water vapour in the tropical lower stratosphere by the upwelling branch of the Brewer-Dobson circulation imprints a seasonal cycle

of $H_2O$ known as the atmospheric tape recorder (Mote et al., 1996) as will be seen in the next section. With increasing altitude,
an increase in $H_2O$ is found to be consistent with the averaged vertical profiles shown in Fig. 3. Higher values of $H_2O$ are found first over high latitudes in the summer hemisphere reflecting the production of WV through methane oxidation under a long duration of sunlight (LeTexier et al., 1988). In general, $H_2O$ shows the zonal mean expected distribution that has been established in previous studies (e.g., Randel et al., 2001).

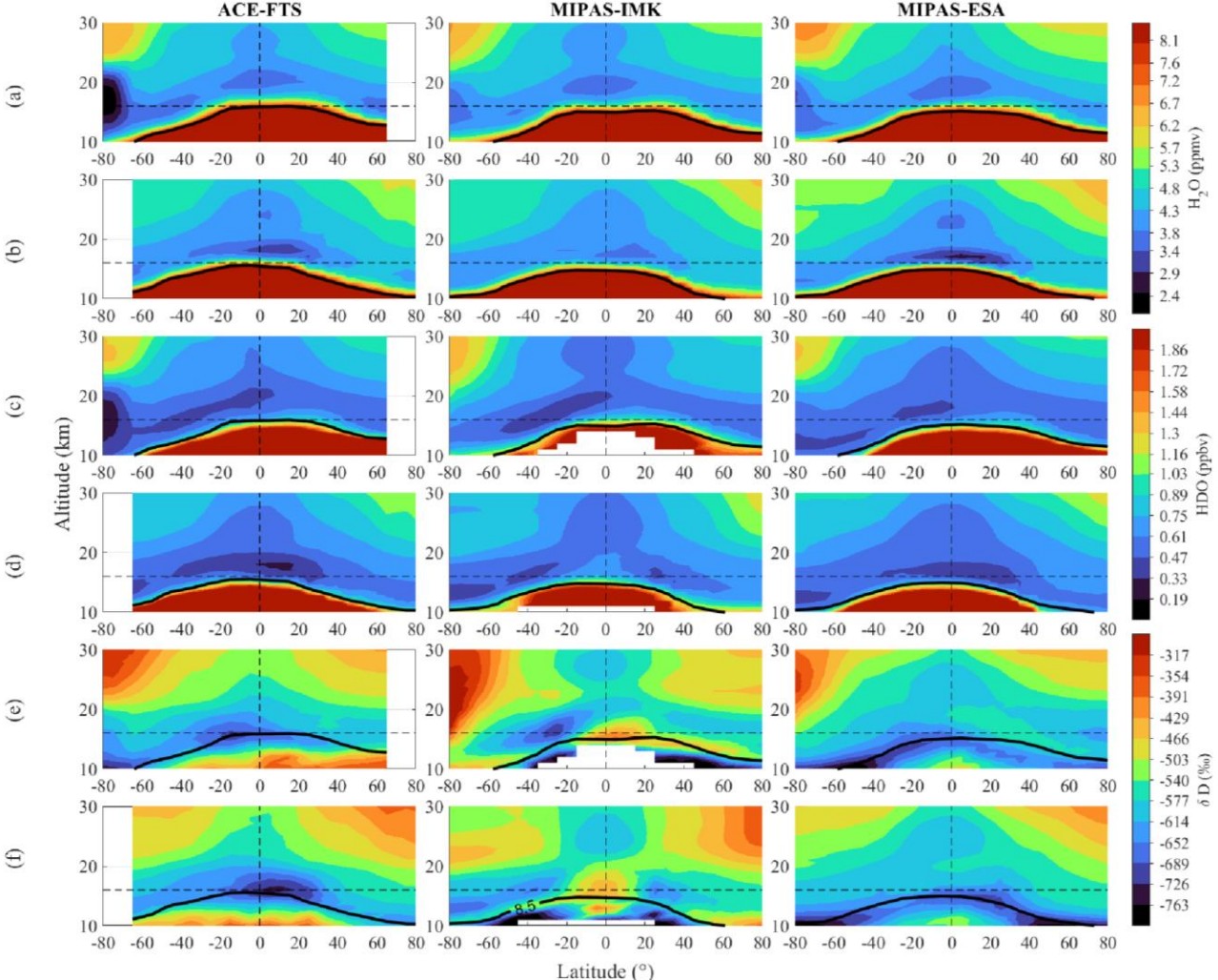

**Figure 5.** Latitude-altitude cross sections of $H_2O$ in (a) boreal summer (JJA) and (b) boreal winter (DJF), HDO for (c) boreal summer and (d) boreal winter and $\delta D$ during I boreal summer and (f) boreal winter for the three datasets. The left column represents ACE-FTS data, the middle column represents the MIPAS-IMK data and the right column the MIPAS-ESA data. The climatology is based on the 2004-2012 period. The absence of profiles in MIPAS- IMK map below the tropical tropopause is due to a more stringent cloud filtering approach used by IMK. Black line indicates the climatological tropopause.

The general distribution of HDO (Fig 5 (c) and 5 (d)) shows some similarities to that of $H_2O$ (Fig. 5 (a) and 5 (b)), reflecting that both species have a common in situ source in the stratosphere, i.e., oxidation of methane and hydrogen. In Antarctica, both $H_2O$ and HDO values in the polar vortex are lower than for the corresponding Artic polar vortex. These lower values evidence the effect of dehydration through the formation of Polar Stratospheric Clouds (PSCs). However, it is worth commenting that for the ACE-FTS data, the minimum values in the Antarctic polar vortex during the JJA are very low (< 3

ppmv for $H_2O$ and < 0.3 ppbv for HDO) compared to the two MIPAS datasets (in the range of 3.4 to 3.8 ppmv for $H_2O$ and 0.3 to 0.5 for HDO ppbv). ACE-FTS does not include data from all the local winter months because of the requirement for sunlight for its measurements. This requirement leads to ACE-FTS values sampling only during the later part of this season (i.e., August vs. June – August) at the highest latitudes, while the two MIPAS data sample during the three months, and this likely leads to ACE-FTS showing more dehydration than MIPAS.

Fig 5 (e) and 5 (f) depict the δD averaged latitude-altitude cross sections for JJA and DJF respectively. Large differences between the three datasets are found in the tropical upper troposphere due to the influence of clouds and limitations of MIPAS measurements in lower altitudes. Large depletion in δD is found on top of the climatological tropopause for MIPAS-ESA and ACE-FTS. The depletion occurs in MIPAS-IMK at a higher altitude, especially above the tropical tropopause. It is known that water vapour transported from the troposphere to the stratosphere is stronger depleted in the heavier isotopologues

while the oxidation of methane in the stratosphere should cause an increase in the isotopic ratio (Wang et al. 2018). The most evident feature at higher altitudes (between roughly 20 and 30 km) is the δD annual cycle with higher values during local summertime and lower values during local wintertime over the high latitudes due to the downwelling of older air which has had more time for methane oxidation (Stiller et al., 2012). This effect is found for the three databases, but there are also differences between them at higher latitudes. In the Antarctic region, the expected asymmetry with latitude driven by the winter

polar vortex due to the influence of PSCs on δD values is observed in ACE-FTS data, but it is absent for MIPAS-IMK data. In the case of MIPAS-ESA, the potential influence of PSCs on δD in the Antarctic region is very subtle for JJA compared with DJF.

        The results obtained with δD for ACE-FTS are in complete agreement with those of Randel et al. (2012) from previous data versions (2004 to 2009). δD for MIPAS-IMK is only partially in agreement with Högberg et al. (2019) since these authors

observed minimum values in the lower stratosphere over the Antarctic polar vortex (75 S to 80 S) during the austral winter in previous version of the data (2002 to 2004). As stated earlier in this work, zonal mean distributions of δD for MIPAS-ESA have never been compared before.

**4.4. Comparison of the tropical seasonal cycle**

        Several details of the vertical propagation of the tropical seasonal signal along the monthly evolution of the three

databases are shown in Fig. 6, which depicts the height-time diagrams covering 30 S and 30 N of $H_2O$ (left panels), HDO (central panels) and δD (right panels) concentrations. Left panels show minimum annual values in $H_2O$ originating near the tropical tropopause and propagating vertically upwards, which is known as the tape recorder signature (Mote et al., 1996). The

overall picture is equivalent for the three data sets, but differences in details are found. ACE-FTS signal is noisier as this dataset has coverage over the tropics typically only for four months (February, April, August, and October). The tape recorder signature is clearly seen but up to 25 km of altitude. The two MIPAS data sets exhibit a stronger tape recorder in terms of its amplitude than the ACE-FTS data. However, for MIPAS-ESA the signal is also larger below 25 km and for MIPAS-IMK the annual variation is found to extend to larger altitudes.

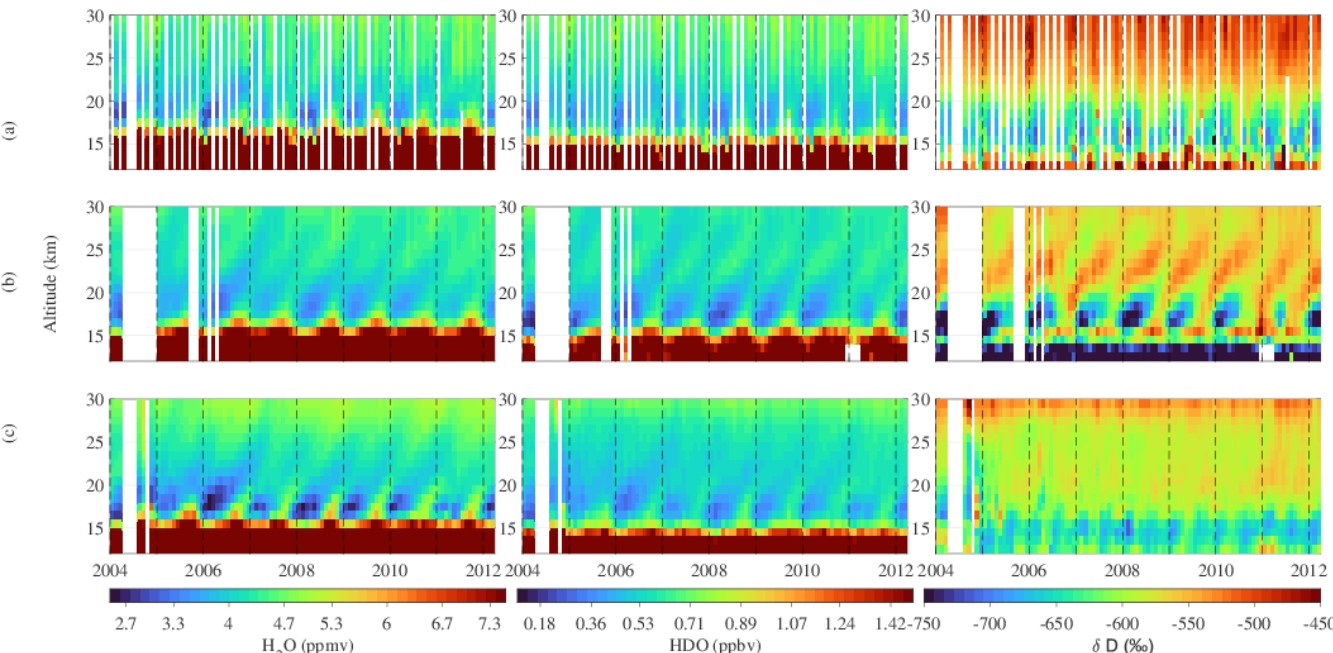

**Figure 6.** Altitude vs. time diagrams over 30 S and 30 N of $H_2O$, HDO and $\delta D$ for the datasets (a) ACE-FTS, (b) MIPAS-IMK, and (c) MIPAS-ESA. Data coverage between MIPAS-ESA and -IMK differs because IMK data are from the nominal observation mode only, which was not operated in Aug 2004, Sep to Nov 2005, and Feb and Apr 2006, while MIPAS-ESA data cover other observation modes as well.

The picture of HDO temporal evolution (central panels) is very similar to the $H_2O$ picture. The exception is that the HDO annual variation in ACE-FTS is found to be weaker and confined to lower levels compared to $H_2O$ annual variation and, by contrast, the tape recorder signature in MIPAS-ESA is extended up to approximately 28 km of altitude and MIPAS-IMK even higher.

Right panels depict altitude-time variation of $\delta D$. At 15 km, above the tropopause, a deuterium depletion over the year (compared to SMOW) is observed with variations within -680 to -600‰ for MIPAS-ESA, -680 to -550‰ for ACE-FTS and -500 to -680‰ for MIPAS-IMK. ACE-FTS data show the characteristic tape recorder pattern of the annual $\delta D$ minimum, although the annual fluctuations in the lower stratosphere are small. MIPAS-ESA show a very weak signature of the tape recorder which seems to be consistent with ACE-FTS result. By contrast, in MIPAS-IMK, the $\delta D$ annual variation related to

the tape recorder signature is evident with a steep gradient between the dry and wet phases in the lower stratosphere. Lossow et al. (2020) showed that a tape recorder signal exists in ACE-FTS V3.5 data as well, although with a lower seasonal amplitude of ~ 25 ‰ in contrast to MIPAS-IMK δD data, that have (in the data version investigated there) a seasonal amplitude of about 75 ‰. Figure 6 demonstrates that the differences in seasonal amplitudes found for older data versions remain for the most recent data versions.

## SUMMARY AND CONCLUSIONS

The stratospheric water vapor (SWV) has a significant climate feedback, which makes quantitative estimates of SWV budget changes necessary. Furthermore, there still remains many uncertainties related to the origin of the SWV (Konopka et al., 2023) and current climate models show substantial biases in the water vapor content of the lowermost stratosphere (Charlesworth et al., 2023). The entry of water vapor into the stratosphere is controlled by chemical and dynamical processes in the lower stratosphere (LS) presenting a challenge for understanding and modelling this region. By adding the isotopic processes in the analytical and numerical models and by comparing modelled and measured isotopic composition in water vapor, model's transport processes can be directly validated. Therefore, accurate stratospheric δD data is of utmost importance to validate water vapor transport studies and to improve biases in the SWV of climate models.

Previous comparisons of δD data in the stratosphere with MIPAS-IMK and ACE-FTS used a very limited period of time. Högberg et al (2019) assessed the profile-to-profile comparisons of stratospheric δD using the overlap period between the two datasets from February 2004 to March 2004. During this short overlap period most of the coincidences are concentrated near 70 N. Lossow et al. (2020) reassessed the discrepancies in the annual variation of δD in the tropical lower stratosphere, but the MIPAS-IMK dataset only covered the period from July 2002 to March 2004. Therefore, longer time series are needed to draw robust conclusions.

This work presents $H_2O$, HDO and δD comparisons among 3 data sets of stratospheric data from two different satellite instruments, ACE-FTS and MIPAS. The recent data versions ACE-FTS V4.1/4.2, MIPAS-IMK V5H_H2O_20, V5R_H2O_220/221, V5H_HDO_22 and V5R_HDO_222/223 and MIPAS-ESA Level 2 V8 were compared. Specifically, the comparison with MIPAS-ESA is performed for the first time in this work for the period 2004 - 2012. The database comparison is based on two approaches: profile-to-profile comparisons and climatology comparisons not requiring coincidences of the observations. The main conclusions of this study are summarized as follows:

The mean profiles of $H_2O$, HDO, and δD between 16 and 30 km, averaged over all latitudes, show remarkable similarity between ACE-FTS and MIPAS datasets, with only minor differences observed within these altitudes. Above 30 km, the $H_2O$ ACE-FTS data show a dry bias, while MIPAS-ESA data show a dry bias for HDO. As a consequence, a negative/positive bias was found for MIPAS-ESA/ACE-FTS δD data upwards 30 km of altitude. Therefore, the climatological analysis was restricted to the range between 16 and 30 km which corresponds to the lower and the middle stratosphere.

Biases from profile-to-profile comparisons exhibited the quantitative differences between the average profiles. Coincident profiles at all latitudes indicate a general good agreement in ACE-FTS comparisons for $H_2O$, HDO and $\delta D$ within $\pm13.4\%$ in the relative bias for the altitude range $16 - 30$ km. For $H_2O$ the better agreement is found between MIPAS-IMK and ACE-FTS with values in the range $-0.05$ to $0.16$ ppmv ($-0.9\%$ to $3.6\%$). However, comparisons between MIPAS-ESA and ACE-FTS show the lower absolute and relative bias both for HDO ($-0.02$ to $0.05$ ppbv and $-1.8$ to $8.7\%$) and $\delta D$ ($-41.2$ to $10.5‰$ and $-8.5$ to $1.8\%$). The $\delta D$ measurements obtained here are comparable to those obtained by Högberg et al. (2019) for previous versions of MIPAS-IMK and ACE-FTS data. Högberg et al. (2019) performed four comparisons between different MIPAS-IMK vs. ACE-FTS versions obtaining biases in $\delta D$ typically within $\pm30‰$ (corresponding to $\pm10\%$ in relative terms) for the lower and middle stratosphere. In this work, similar biases are found within the same altitude range. Furthermore, our results are considerably more robust than those of Högberg et al. (2019) because of the limited period of time analysed by these authors (from the second half of February 2004 to end of March 2004), with the number of coincident profiles varying between 300 and 400. Our comparisons are typically based on several thousand coincidences during a time period of 9 years. Furthermore, our results are complemented by the comparisons with new MIPAS-ESA data, which indicate for $\delta D$ even a better agreement with ACE-FTS than MIPAS-IMK - ACE-FTS agreement.

We also analysed latitude-altitude cross sections considering all measurements of the datasets in the latitude range from 80 S to 80 N. Consistent with previous observations (Randel et al., 2012; Högberg et al., 2019), the overall vertical structure of $H_2O$, HDO and $\delta D$ exhibits a large depletion near the tropopause, and higher mixing ratios between 20 and 30 km over the poles during the local summertime because of the methane oxidation. However, there are also some differences between the results of each dataset. The tropical depletion of $\delta D$ in ACE-FTS and MIPAS-ESA occur on the top of the dynamical tropopause, but the minimum is found at higher altitudes in the MIPAS-IMK dataset. Large differences are also found between the two MIPAS data sets over the tropical upper troposphere, probably related to a different approach used by the two MIPAS algorithms to handle cloud contamination. In agreement with Hogberg et al (2019) and because ACE-FTS instrument measures at lower altitudes, it can be concluded that ACE-FTS data are probably more realistic at these altitudes. Regarding the Antarctic region, ACE-FTS shows lower $\delta D$ values over the polar vortex than the MIPAS datasets, likely related to PSCs. Nevertheless, the ACE-FTS lower values can be partially attributed to sampling error as ACE-FTS data only cover a 15-days period during the late winter. These results are not representative of the 3-month season mean of MIPAS measurements, which also includes the first months of the winter when the PSCs areal coverage has not yet peaked. MIPAS-ESA barely shows $\delta D$ minimum values over the Antarctic polar vortex and MIPAS-IMK data do not show them over the highest latitudes. Latitude-altitude sections of $\delta D$ for MIPAS-ESA have never been shown before.

Finally, the general depiction of the tape recorder signal in $H_2O$ and HDO for the three databases seems to be reasonable. However, the temporal variations of $\delta D$ in the lower and middle stratosphere show larger discrepancies. The annual variation for ACE-FTS data and MIPAS-ESA data is very weak compared to the MIPAS-IMK dataset, which shows a coherent tape recorder signal clearly detectable up to at least 30 km. Lossow et al (2020) showed a similar result with previous versions of MIPAS-IMK and ACE-FTS data. They performed some tests to reveal the main reason for the differences in the annual

variation of δD. They found that while the differences in the temporal sampling between the MIPAS-IMK and ACE-FTS data sets are not the main reason for the differences in the annual variation of δD at least in the lowermost stratosphere, some issues related to the quality of the MIPAS $H_2O$ data used in this context, and the differences in vertical resolution between $H_2O$ and HDO potentially contributed to the δD tape recorder differences between MIPAS-IMK and ACE-FTS. This issue remains open.

Considering that MIPAS and ACE-FTS are the only instruments so far which have measured or are measuring both $H_2O$ and HDO simultaneously from satellite on a long period, further improvements in the data sets are highly welcome to understand and reduce the differences in the zonal mean distributions and the annual variation of δD. With this knowledge, the representation of stratospheric water vapor in models would be improved offering promising prospects for future research.

**Code availability**

The scripts for data extraction, profile-to-profile comparison, and climatological analysis in MATLAB is available from the authors upon request.

**Data availability**

The MIPAS-IMK $H_2O$ and HDO datasets can be accessed from the website of the Institute of Meteorology and Climate Research - Atmospheric Trace Gases and Remote Sensing (IMKASF) at https://www.imk-asf.kit.edu/english/308.php. The ACE-FTS data can be accessed and downloaded from the website https://databace.scisat.ca/level2. The MIPAS-ESA data is available online and can be downloaded from the FTP server ftp://mip-ftp-ds.eo.esa.int/ using an FTP client.

**Author contribution:**

L.A. P.O and G.P.S. conceived and designed the research. K.DLR. developed the analysis. P.O and K.DLR. prepared the manuscript draft. K.DLR., P.O., G.P.S. M.K., P.R., M.G., K.A.W., C.P-O. and L.A. reviewed and edited the manuscript. All authors have read and agreed to the published version of the manuscript.

**Competing interests:**

One of the authors (G.P.S.) is associate editor of AMT.

**Acknowledgements**

This research has been supported by the following grants: the Spanish Ministerio de Economía y Competitividad (grant no. CGL2016-78562-P), PAPIIT (DGAPA-UNAM) IN116120, IG101423 and CONACyT 315839. Paulina Ordoñez is grateful for the support of Maria Zambrano (UPO; Ministry of Universities; Recovery, Transformation and Resilience Plan -Funded by the European Union -Next Generation EU). The Atmospheric Chemistry Experiment (ACE) is a Canadian-led mission mainly supported by the CSA and the NSERC, and Peter Bernath is the principal investigator. The IMK team would like to thank the European Space Agency for making the MIPAS level-1b data set available. We acknowledge Michael Kiefer for his assistance with IMK data management and providing comments during the early phase of the manuscript. Karen de los Rios is grateful to the National Council on Science and Technology (CONACYT) for their generous financial support and scholarship. Special appreciation is extended to J. R. Torres-Castillo for his invaluable assistance in enhancing the employed algorithms. Additionally, Karen de los Rios acknowledges R. Stanley Molina-Garza for his insightful recommendations.

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
