# Peer review of "Comparison of the $H_2O$ , HDO and $\delta D$ stratospheric climatologies between the MIPAS-ESA v8, MIPAS-IMK v5 and ACE-FTS V4.1/4.2 satellite data sets"

_EGUsphere, 2023_

## Author Response (AR1)

**Revision of "Comparison of the H2O, HDO and δD stratospheric climatologies between the MIPAS-ESA v8, MIPAS-IMK v5 and ACE-FTS V4.1/4.2 satellite data sets" by De los Rios et al., submitted to AMT.**

Dear Editor,

We have now completed substantial revisions of our manuscript submitted to AMT. We reply below to the detailed comments by the two reviewers. We have addressed all points, and in addition made ample edits to address them, as well we added several references.

In our reply below, we show the reviewers comments in black colour, the response to the reviewer is in blue colour and the action performed in the revision is indicated in red colour.

With best regards, on behalf of all authors,
Paulina Ordoñez

**Response to Referee #1**

General Comments

We thank the reviewer for his/her constructive comments, which will result in an improvement of the manuscript.

1. This paper compares MIPAS and ACE $H_2O$ and HDO over the period 2004 to 2012 when MIPAS was working. Similar comparisons have been done previously, such as Lossow et al., 2020, Lossow et al., 2011, Sheese et al., 2017, Ordonez-Perez et al., 2021, Risi et al., 2011, Hogberg 2019. This latest comparison utilizes more recent versions of the data products which are presumably better. Indeed, the authors state "The HDO data version used here differs significantly from the data versions assessed by Lossow et al. (2020) and Högberg et al. (2019) and used by Steinwagner et al. (2007, 2010)". In this latest paper, a new MIPAS product is presented for the first time: MIPAS-ESA. This and MIPAS-IMK are compared with ACE and with each other. It is not clear to me where, in the processing chain, these two MIPAS products diverge. The retrieval methods seem to be different. Not clear whether the spectra are the same.

The two products differ already in the version of the spectral data. While the ESA product uses Level-1b version 8.03, the IMK data uses version 5.02/5.06. The differences between the versions result from several factors including improved calibration procedures, a compensation of the detector drift due to aging, and subtle adjustments of the geolocations. We are aware that it is not optimal to use different level-1b data versions, however, it is unavoidable in this case, since there are no earlier HDO data versions of the ESA product, and HDO from version 8 is not yet available from the IMK processing.

The level-2 processing (retrieval of trace gases) has many substantial differences between the processors: in general, MIPAS IMK and MIPAS ESA use different spectral intervals, and there is a rich literature about the differences of the retrieval set ups and the reasons for these choices. In particular, the two papers Laeng et al., 2015, and Raspollini et al., 2013 describe differences between the algorithms performing MIPAS analysis and the differences in the products for $O_3$ and other trace species, but not HDO. For additional papers on the MIPAS IMK data set and retrievals, we refer one to the web site on the IMK products (https://www.imk-asf.kit.edu/english/298.php), while the evolution of MIPAS ESA Level 2 algorithm and its products is described in these papers (Ridolfi et al., 2000; Raspollini et al., 2006, Raspollini et al., 2013, Raspollini et al., 2022, Dinelli et al., 2021) and references therein; it is beyond the scope of this paper to summarize them all.

We will clarify this in the revised version of the manuscript.

Section 2 has been edited to better clarified the differences between MIPAS-IMK and MIPAS-ESA.

2. To be honest, I didn't feel that I learned much reading this paper. There have already been several similar comparisons using earlier versions of the MIPAS and ACE data products.

It is true that several comparisons have already been made. Recently, the SPARC-WAVAS-II activity compared $H_2O$ and HDO from all available satellite instruments since the year 2000, also including ACE-FTS version 3.5, MIPAS IMK version 5, and ESA version 5 and 7 (only for $H_2O$) (see ACP/AMT Special Issue "Water vapour in the upper troposphere and middle atmosphere: a WCRP/SPARC satellite data quality assessment including biases, variability, and drifts", https://amt.copernicus.org/articles/special_issue10_830.html). Newer data versions of $H_2O$ and HDO have been used in our paper here for all three data sets. We think that it is useful to assess the quality of new data products from satellite data, in particular here the new MIPAS ESA product, and it is necessary to do so before further scientific work with these data can be done.

Particularly important is the case of using δD data to study the origin of water vapor that enters the stratosphere. Therefore, it is critical to understand the quality of the $H_2O$ and HDO that are used in this calculation as is a major focus of this paper.

With the only exception of MIPAS-IMK $H_2O$ data, which use MIPAS-IMK V5H_H2O_20 (2002-2004) and V5R_H2O_220/221 (2005-2012) as in Lossow et al (2019), we employ newer data sets than those used in the previous studies as now indicated in lines 109-114.

This importance of publishing new data version comparisons has been highlighted in lines 21 – 22. The necessity of accurate δD data in the stratosphere is described in lines 494-501 of the revised version of the paper.

3. The authors show that the MIPAS-IMK $H_2O$ and HDO products agree well with ACE, but the MIPAS-ESA HDO profiles are discrepant around 40 km altitude. Since the error bars on the MIPAS HDO profiles are quite large above 30 km, the discrepancy is not significant.

Line 21 states "Stratospheric $H_2O$ and HDO global average coincident profiles reveal good agreement." I disagree. In my opinion a 37.5% bias in HDO at 40 km is not good agreement. Although the MIPAS HDO profiles have large enough uncertainties such that they bridge this 37.5% gap, this doesn't mean that the agreement is good. It just means that the MIPAS HDO measurements are not useful at 40 km and above.

We agree with this comment. In addition, we note that ACE-FTS $H_2O$ has a significant deviation from the other two data records. This deviation was also found in the SPARC/WAVAS-II comparisons for earlier data versions (see, for example, Lossow et al., 2019) with respect to many other satellite data records. Therefore, because of these known discrepancies, we will restrict our extended analyses (those after discussion of Fig. 3) to altitudes below 30 km in the revised version of the paper.

Our climatological analysis is restricted to the range between 16 and 30km is indicated in lines 331-332 and 517-518 of the revised version of the paper.

It is worthy comment that in the new version of the manuscript we report the standard error of the mean (SEM) instead of the standard error as indicated in line 256-257. This is because the standard error gives an indication on the precision of the single measurements. In Fig. 3 we computed the averages of many profiles. Due to the large number of data points, the comparisons are highly significant, and we are interested in their biases.

Specific Comments

4. Lines 24-25 of the abstract state "ACE-FTS agrees better to MIPAS-IMK than MIPAS-ESA, with biases of -4.8% and -37.5%, respectively. The HDO bias between MIPAS-IMK and MIPAS ESA is 28.1 % at this altitude" So ACE is 4.8% lower than MIPAS-IMK and 37.5% lower than MIPAS-ESA. One might naively expect MIPAS-IMK HDO to be 37.5-4.8 = 32.7% lower than MIPAS-ESA. But it is only 28.1% lower. Presumably this is because different data were used for comparing ACE with MIPAS, than comparing MIPAS-IMK and MIPAS-ESA. Perhaps this should be made clearer in the text.

Indeed, a different number of coincident profiles were used in each of the three comparisons. There were differences in the geolocations because the values were arithmetically averaged if multiple coincidences were found in each of the coincidence regions, as we wrote in the text. However, we concur with the referee in the necessity of compare data with the same geolocation.

In the revised version of the manuscript, when multiple MIPAS profiles are spatially coincident with an ACE-FTS profile, the MIPAS profile closest in time is selected. In addition, there must be both MIPAS IMK and MIPAS ESA processed data available for this coincident profile.

The new procedure for selecting coincident profiles is described in lines 230-238. Please, see also lines 241-242, where we state that only points with triple observation are used.

5. Lines 29-30 of the abstract state: " ...aligns more closely with expected stratospheric behavior for the entire stratosphere". Delete "stratospheric" or " for the entire stratosphere". This is unnecessary to have both.

Will be done.

The sentence has been deleted since δD data have not been validated before, and thus we don't have enough evidence about if the stratospheric behaviour of δD really aligns more closely with the MIPAS-IMK results.

6. Also, this sentence states that MIPAS-IMK calculates dD. I consider it more of a measurement. A model would calculate dD.

Will be adjusted in the paper.

Thank you again for the comment. This sentence has been deleted.

7. Line 15: I've not seen the word "isotopological" before. According to Google it is a mathematical term meaning "having the same topology". Perhaps the authors mean "isotopic"? Line 29 of the abstract uses "isotopic" in a similar context. The word "isotopological" occurs later in the paper, e.g., lines 54, 56. So I'm not sure if the authors are trying to make a distinction between "isotopological" and "isotopic", or they consider these terms synonymous. I suggest that "isotopological" NOT be used, because mathematicians have already defined this word for use in topology.
8. Line 54 states: "isotopological composition of WV" change to "isotopic composition of WV".
9. Line 56 states: "Among the isotopological species of WV..." Change to "Among the isotopologues of WV...".

Thank you very much for the constructive comment. We were making a distinction between "isotopological" and "isotopic" since "isotopic" is the adjective for "isotope" and "isotopological" is the adjective for "isotopologue".

Certainly, two terms - "isotopologic" (e.g., Herbin et al., 2007; Bahr and Wolff, 2022) and "isotopological" (e.g., Schneider et al., 2020; Israel, 2023) - can be found in the literature to describe characteristics or properties related to "isotopologues". Since the term "isotopological" is also a mathematical term meaning "having the same topology", we think that the suggestion of the referee in not using the term "isotopological" is right, and term "isotopologic" will be used in the revised version of the manuscript.

We use the term "isotopologic" instead of "isotopological" in the revised version of the manuscript (lines 15, 53).

10. Having read the paper, what I would really like to know is why MIPAS-IMK and MIPAS-ESA HDO are different. Presumably these products come from the same raw data. It undermines confidence in MIPAS to see different groups obtain such different results.

We respectfully disagree. It is correct that the MIPAS ESA and MIPAS IMK HDO product comes from the same raw data (interferograms), but the spectral data are from different level-1b data versions. The level-2 processing (the retrieval set-up in this case) is a most relevant part in the data generation. Even the same level-2 processor produces different results with different retrieval settings. In cases where several level-2 processors of other satellite data exist, they often result in differing products (e.g., GOMOS, SCIAMACHY, SMILES, OMPS). The level-1b data for the Envisat instruments were made public to encourage different processing techniques to be developed and applied.

Further, we would like to point out that the differences between the two MIPAS products are rather limited (see Fig. 4). $H_2O$ differences are close to 5% at max, while HDO differences remain <10% (all below 30 km). Differences as large as this can occur between different versions of the same data product of the same processor.

The main differences between the two MIPAS datasets are now described in section 2 of the revised version of the paper.

11. Example of duplication; Line 25-26: The meridional cross-sections of H2O and HDO exhibit the expected distribution that has been established in previous studies. Lines 27-28: The meridional cross-sections of δD are in good agreement with the previous version of MIPAS-IMK and ACE-FTS data. The sentence on lines 27-28 seems superfluous. Given that H2O and HDO are in good agreement with previous datasets and studies, readers will assume that δD will also be in good agreement. No need to tell them that it is.

As the δD here is calculated from individual profiles then averaged rather than mean profiles, it is important to point out this good agreement. From our experience, even subtle differences in $H_2O$ and HDO (such as those between versions) are enlarged severely by calculating δD from them.

The first sentence "The meridional cross-sections of $H_2O$ and HDO exhibit the expected distribution that has been established in previous studies" is still in line 24-25. The second sentence in line 27-28 "The meridional cross-sections of δD are in good agreement with the previous version of MIPAS-IMK and ACE-FTS data" has been deleted since some minor differences have been found between the latitude-altitude cross section computed in this work and previous studies.

In the revised version of the paper, δD has been computed by using the same approach than Högberg et al. (2019) and Lossow et al (2020). We were calculating δD from individual HDO and $H_2O$ profiles and average the results. As indicated now in lines 256-260, we now first compute the averages values of $H_2O$ and HDO from all the profiles and these results are combined to δD. As shown by Högberg et al. (2019) there are some differences between the different approaches to calculate δD results because of their noncommutativity.

12. Raspollini et al., 2022 is cited on lines 86 and 148, but doesn't exist in the References. Either these citations are typos (should be 2020, perhaps?), or the Raspollini 2022 reference is missing.

The reviewer is right, the reference Raspollini et al., 2022 is missing in the list of references, while Raspollini et al., 2020 is correctly reported in the list. We also updated the DOI of reference Dinelli et al., 2021.

Dinelli et al. (2021) is updated in lines 643-646 and the reference Raspollini et al. (2022) is included in lines 813-817 of the revised version of the manuscript.

13. Line 106 states "we focus here on newer data versions that cover the full mission period of ten years.". If the newer data versions cover 10 years, why do all the tables and figures cover only eight years (2004-2012)? Also, this sentence is missing a final ".".

The reviewer is right. The sentence is corrected in the revised version of the manuscript since we are focusing on the overlap period between MIPAS and ACE-FTS which is from 2004 to 2012.

Modified in line 101-102 of the revised version of the manuscript.

14. Lines 152 to 154: It seems that for the MIPAS-ESA processing, different retrieval methods were employed for $H_2O$ and HDO. The text needs to explain why this was necessary. Also, why is it "opportune" to use an a priori atmospheric HDO profile that is $(3.107 \times 10^{-4})$ of that of $H_2O$. This is the value in VSMOW, not the atmosphere. In the UTLS the HDO/$H_2O$ ratio is closer to $(1 \times 10^{-4})$ so using the $3.107 \times 10^{-4}$ value might adversely bias the HDO retrievals.

A different retrieval approach has been used for $H_2O$ and HDO because the approach used for $H_2O$ (namely a Levenberg-Marquardt regularization approach within the iterations followed by an a posteriori regularization) was not sufficient to constrain the HDO retrieval. For the HDO, an a priori error of 100% is used in order not to introduce a bias, as written in the text (line 161).

Clarified in lines 132-147 of the revised version of the manuscript.

15. Tables 1 and 2 can be put side-by-side and hence merged into a single table.

Thank you for the recommendation. The tables will be merged.

The merged table is in line 401 of the revised version of the manuscript.

16. Figure 1. Why are the ACE/MIPAS-IMK coincidences ~5 deg. to the South of the ACE/MIPAS-ESA coincidences? So, there is no overlap in the ACE data used for MIPAS-IMK validation and for MIPAS-ESA validation -- they are at different latitudes and hence dates. I don't understand why the same ACE data can't be used for both.

Thank you for the comment. The referee is right, there were differences in the geolocations of the same profiles in the two data versions due to the method we used for determining the coincident profiles (see comment #4). Now the same ACE-FTS data have been used for both comparisons.

In lines 230-238 of the revised version of the paper, the new procedure to select coincident profiles is described. Please, see also lines 241-242 where we clarify that only points with triple observation are used.

17. Figure 2 should have 1 panel with 3 curves in different colors showing the number of coincidences between: (1) MIPAS-IMK and ACE, (2) MIPAS-ESA and ACE, and (3) MIPAS-IMK and MIPAS-ESA. This will provide the reader more information in less space.

Done in the revised version of the manuscript.

Please, see Figure 2 in line 252 of the revised version of the manuscript.

18. Figure 3: I don't understand the rationale for comparing ACE separately to each MIPAS version. This requires 4 panels and repeats the ACE profiles. Why not have two panels; one for $H_2O$ and the other for HDO? Each panel contains the 3 profiles (ACE, MIPAS-IMK, MIPAS-ESA) in different colors. I guess the reason is that ACE

data compared with MIPAS-IMK is different from that compared with MIPAS-ESA. In which case you need 4 profiles in each panel: MIPAS-IMK, MIPAS-ESA, $ACE_{IMK}$, $ACE_{ESA}$.

We agree with the reviewer suggestion, the three curves will be inserted in one panel. We also improve the figure 3 including the MIPAS-IMK to MIPAS-ESA comparison.

Please, see Figure 3 in line 333 of the revised version of the manuscript.

19. Figure 6 should be appended to the bottom of fig.5, making a single figure with a single caption. This will allow the reader to compare the features in the dD panels with those in the H2O and HDO panels. This won't be possible with the dD panels on a different page. it will also eliminate repetition in the caption.

Thank you for the recommendation. This is done in the revised version of the manuscript.

Please, see Figure 5 in line 429 of the revised version of the manuscript.

20. Similarly, fig. 8 should be appended to the right of fig.7. It has exactly the same x- and y-axes.

Yes, done in the revised version of the manuscript. Thank you for the suggestion.

Please, see Figure 6 in line 473 of the revised version of the manuscript.

21. Line 465 states: " the MIPAS instrument shown a negative bias at the troposphere" Change to " the MIPAS instrument shows a negative bias at the troposphere".

Done in the revised version of the manuscript.

In the new version of the manuscript, we restrict the climatological analysis to the range between 16 km and 30 km (the lower and middle stratosphere). Please see lines 331-332 Therefore, this sentence about bias at the troposphere in the altitude-time diagrams has been removed.

22. Line 420 states:" The general distribution of HDO (Figs 5(c) and 5(d)) shows some similarities to that of H2O (Fig. 5(a) and 5(b)), reflecting that both species have a common in situ source in the stratosphere, i.e., oxidation of CH4 and H2." But HDO comes from CH3D and HD, so change sentence to: "The general distribution of HDO (Fig.s 5(c) and 5(d)) shows some similarities to that of H2O (Fig. 5(a) and 5(b)), reflecting that both species have a common in situ source in the stratosphere, i.e., oxidation of methane and hydrogen."

Done in the revised version of the manuscript.

Modified in line 435-436 of the revised version of the paper.

23. Line 447 states: "...diagrams over 30S and 30N...". This is ambiguous. Perhaps "...diagrams covering 30S to 30N..."

Done in the revised version of the manuscript.

Modified in line 465 of the revised version of the paper.

24. Line 465:"As it was previously shown in the Fig.3, the MIPAS instrument shown a negative bias at the troposphere" Three grammatical errors in this half sentence. Change to: " As was previously shown in Fig.3, the MIPAS instrument shows a negative bias at the troposphere".

Done in the revised version of the manuscript.

This sentence was referred to the troposphere and due to biases in this region, our climatological analysis is now focused in the lower and middle stratosphere. Therefore, this sentence has been deleted.

25. Also, I don't see anything negative in Fig.3. Perhaps the authors mean Fig.4?

Thank you for the observation. Yes, it is figure 4, it is changed in the revised version of the manuscript.

Again, as responded in the previous comment (#24), this sentence has been removed.

26. Line 490 states: " The analysis conducted in this study highlights a higher level of agreement in HDO measurements obtained from ACE-FTS in both comparison cases." This seems to imply that ACE agrees better with MIPAS than MIPAS-IMK and MIPAS-ESA agree with each other?.

It was not our intention to imply this. In fact, the agreement between the two MIPAS products of $H_2O$ is at least as good (below 30 km), as that between ACE-FTS and one of the MIPAS products. Regarding HDO, the differences between the two MIPAS data sets are somewhat larger, indeed, especially between 20 and 30 km, but still < 10%.

This paragraph has been reworded, please see lines 340-351 and 363-371 of the revised version of the paper.

The better agreement is found in comparisons to ACE-FTS both for $H_2O$ and HDO. Lower biases are found for the MIPAS-IMK – ACE-FTS comparisons in $H_2O$ and for MIPAS-ESA and ACE-FTS comparisons in HDO. Differences between the two MIPAS are not much higher, but they are discussed in these paragraphs.

27. Line 524 states: "the findings from this study suggest that the MIPAS-IMK dataset provides a more realistic signal for the entire stratosphere". More realistic than what? ACE or MIPAS-ESA.

"More realistic than MIPAS-ESA" was to be meant. However, we will reword the sentence in the revised version of the manuscript since the affirmation "more realistic" is inaccurate given the existing dD data.

This sentence has been deleted in the revised version of the paper.

28. Line 526 states: "it is crucial to exercise caution when interpreting these results, specifically considering the sampling limitations of ACE-FTS in the tropics, during the period of study, especially at lower altitudes." I don't recall much discussion of this in the main part of the paper. It is true that ACE occultations are sparse in the tropics and that high clouds can often limit penetration of the troposphere. But it seems unfair to ACE to call this a conclusion. And it not clear what altitude range this comment is aimed.

Thank you for the comment, we concur with the reviewer that this is not a conclusion of this analysis. The last paragraph of the revised version of manuscript will be modified.

This paragraph has been removed in the revised version of the paper.

29. Line 530: "The code in MATLAB is available from the authors upon request." The term "the code" is too vague. Add one sentence explaining what "the code" does.

Changed in the revised version of the manuscript.

The code includes scripts for data extraction, for profile-to-profile comparison, and for climatological analysis as indicated in line 565.

30. The format of the References is unfriendly. There is no indentation at the start of a new reference, nor a gap between references. So, it is hard to tell where one reference ends and the next begins. Perhaps this is the journal style.

Thank you for the observation. Changed.

Please, see the References section in line 596 of the revised version of the paper.

References:

Israel, F. P. (2023). Central molecular zones in galaxies: Multitransition survey of dense gas tracers HCN, HNC, and HCO+. Astronomy and Astrophysics, 671, A59.

Laeng, A.; Hubert, D.; Verhoelst, T.; von Clarmann, T.; Dinelli, B. M.; Dudhia, A.; Raspollini, P.; Stiller, G.; Grabowski, U.; Keppens, A.; Kiefer, M.; Sofieva, V.; Froidevaux, L.; Walker, K. A.; Lambert, J. -C.; Zehner, C., The ozone climate change initiative: Comparison of four Level-2 processors for the Michelson Interferometer for Passive Atmospheric Sounding (MIPAS), REMOTE SENSING OF ENVIRONMENT, https://doi.org/10.1016/j.rse.2014.12.013, 2015

Lossow, S., Khosrawi, F., Kiefer, M., Walker, K. A., Bertaux, J.-L., Blanot, L., Russell, J. M., Remsberg, E. E., Gille, J. C., Sugita, T., Sioris, C. E., Dinelli, B. M., Papandrea, E.,

Raspollini, P., García-Comas, M., Stiller, G. P., von Clarmann, T., Dudhia, A., Read, W. G., Nedoluha, G. E., Damadeo, R. P., Zawodny, J. M., Weigel, K., Rozanov, A., Azam, F., Bramstedt, K., Noël, S., Burrows, J. P., Sagawa, H., Kasai, Y., Urban, J., Eriksson, P., Murtagh, D. P., Hervig, M. E., Högberg, C., Hurst, D. F., and Rosenlof, K. H.: The SPARC water vapour assessment II: profile-to-profile comparisons of stratospheric and lower mesospheric water vapour data sets obtained from satellites, Atmos. Meas. Tech., 12, 2693–2732, https://doi.org/10.5194/amt-12-2693-2019, 2019.

Raspollini, Piera; Arnone, Enrico; Barbara, Flavio; Carli, Bruno; Castelli, Elisa; Ceccherini, Simone; Dinelli, Bianca Maria; Dudhia, Anu; Kiefer, Michael; Papandrea, Enzo; Ridolfi, Marco, Comparison of the MIPAS products obtained by four different level 2 processors, ANNALS OF GEOPHYSICS, https://doi.org/10.4401/ag-6338, 2013

Raspollini, P., Belotti, C., Burgess, A., Carli, B., Carlotti, M., Ceccherini, S., Dinelli, B. M., Dudhia, A., Flaud, J.-M., Funke, B., Höpfner, M., López-Puertas, M., Payne, V., Piccolo, C., Remedios, J. J., Ridolfi, M., and Spang, R.: MIPAS level 2 operational analysis, Atmos. Chem. Phys., 6, 5605–5630, https://doi.org/10.5194/acp-6-5605-2006, 2006.

Raspollini, P., Carli, B., Carlotti, M., Ceccherini, S., Dehn, A., Dinelli, B. M., Dudhia, A., Flaud, J.-M., López-Puertas, M., Niro, F., Remedios, J. J., Ridolfi, M., Sembhi, H., Sgheri, L., and von Clarmann, T.: Ten years of MIPAS measurements with ESA Level 2 processor V6 – Part 1: Retrieval algorithm and diagnostics of the products, Atmos. Meas. Tech., 6, 2419–2439, https://doi.org/10.5194/amt-6-2419-2013, 2013

M. Ridolfi, B. Carli, M. Carlotti, T. von Clarmann, B. M. Dinelli, A. Dudhia, J.-M. Flaud, M. Höpfner, P. E. Morris, P. Raspollini, G. Stiller, and R. J. Wells, "Optimized forward model and retrieval scheme for MIPAS near-real-time data processing", Applied Optics, 39, 1323-1340 (2000).

Schneider, A., Borsdorff, T., Aemisegger, F., Feist, D. G., Kivi, R., Hase, F., ... & Landgraf, J. (2020). First data set of $H_2O$/HDO columns from the Tropospheric Monitoring Instrument (TROPOMI). Atmospheric Measurement Techniques, 13(1), 85-100.

Zeng, Z. C., Addington, O., Pongetti, T., Herman, R. L., Sung, K., Newman, S., ... & Sander, S. P. (2022). Remote sensing of atmospheric HDO/H2O in southern California from CLARS-FTS. Journal of Quantitative Spectroscopy and Radiative Transfer, 288, 108254.

Zeng, Z. C., Addington, O., Pongetti, T., Herman, R. L., Sung, K., Newman, S., ... & Sander, S. P. (2022). Remote sensing of atmospheric HDO/H2O in southern California from CLARS-FTS. Journal of Quantitative Spectroscopy and Radiative Transfer, 288, 108254.

Lossow, S., Steinwagner, J., Urban, J., Dupuy, E., Boone, C. D., Kellmann, S., ... & Stiller, G. P. (2011). Comparison of HDO measurements from Envisat/MIPAS with observations by Odin/SMR and SCISAT/ACE-FTS. Atmospheric Measurement Techniques, 4(9), 1855-1874.

Lossow, S., Högberg, C., Khosrawi, F., Stiller, G. P., Bauer, R., Walker, K. A., ...and Eichinger, R. (2020). A reassessment of the discrepancies in the annual variation of δD-H 2

O in the tropical lower stratosphere between the MIPAS and ACE-FTS satellite data sets. Atmospheric Measurement Techniques, 13(1), 287-308.

Raspollini, P., Arnone, E., Barbara, F., Bianchini, M., Carli, B., Ceccherini, S., Chipperfield, M. P., Dehn, A., Della Fera, S., Dinelli, B. M., Dudhia, A., Flaud, J.-M., Gai, M., Kiefer, M., López-Puertas, M., Moore, D. P., Piro, A., Remedios, J. J., Ridolfi, M., Sembhi, H., Sgheri, L., and Zoppetti, N.: Level 2 processor and auxiliary data for ESA Version 8 final full mission analysis of MIPAS measurements on ENVISAT, Atmos. Meas. Tech., 15, 1871–1901, https://doi.org/10.5194/amt-15-1871-2022, 2022.

**Response to Referee #2**

**Overview**

We thank the reviewer for his/her comments, which will result in an improvement of the manuscript.

The paper compares the H2O and HDO data retrieved from the ACE-FTS solar occultation instrument and from two different retrieval algorithms applied to the MIPAS limb-emission instrument. This is an update on similar work performed by Hogberg and Lossow in 2019 but using reprocessed ACE-FTS and MIPAS-ESA data. However, it is difficult to know what conclusions can be reached, and how or if these have changed with the new data.

In addition to the new ACE-FTS and MIPAS-ESA versions, the MIPAS-IMK data are a new version and employ a new processor approach, too. We say this explicitly in line 96 and specify which version we use in line 118. In the revised paper, we will make clearer which version was used in previous papers, and which version we use here.

With the only exception of MIPAS-IMK H2O data, which use MIPAS-IMK V5H_H2O_20 (2002-2004) and V5R_H2O_220/221 (2005-2012) as in Lossow et al. (2019), we employ newer data sets than those used in the previous studies as now indicated in lines 109-114.

The data used by Högberg et al (2019) are indicated in lines 91-95 of the revised version of the manuscript. The data used by Lossow et al (2019) and Lossow et al (2020) are made clear in lines 85-88 and 95-99 respectively.

This seems a rather 'mechanical' paper - mostly just reproducing earlier results, the only new aspect being the updated datasets. Indeed, the sort of paper one can imagine being generated, in a few years if not already, by some of the more advanced AI machines.

It might have been acceptable if the original authors wanted to update their paper with new results, in which case I would expect a narrative focusing on the algorithm changes, and the expected and observed impacts on the intercomparisons with respect to their earlier results rather than, as here, analysing the results in absolute terms as if they were being presented for the first time. But if it's a paper with new lead authors it also needs some significant new insight or analysis. I also have some doubts about the methodology.

It is critical to intercompare and validate each new satellite dataset as it is produced. As we have new data versions and methodologies, the data users need to understand changes and updates to these $H_2O$ and HDO data products. This validation work should not only fall to the data producers but should be taken up by other members of the community.

In case of MIPAS-IMK (versus ACE-FTS), we refer several times to previous results (e.g., line 315/316; line 335-337; line 373-377; 436-438; 458-461; 492/493; 501-503; 513-515). We will make clearer in the revised version what changes in the retrieval set-up were made and will discuss the expected versus observed changes between the data versions.

Furthermore, we would like to point out that it was by intention that we followed the methodology of Högberg et al. so closely. By using the same methods, we ensure that the new results of this paper are directly comparable to the results presented in Högberg.

Comparisons with Högberg et al. (2019) can be found now in lines 324-327, 381-384, 459-461, 524-532, 534-536.

**General Comments and Suggestions**

1) Colocations

The comparison has been performed on the two versions of MIPAS data as if these were independent satellites. A more satisfactory approach would have been, first, to apply the respective filters to these MIPAS datasets and take just the common profiles, and then compare this with ACE-FTS. This would not only ensure no time/space mismatch between the two MIPAS datasets but also ensure exactly the same time/space mismatch between ACE-FTS and either of the MIPAS datasets.

On this topic, Fig 1 looks very odd. It seems most unlikely that in a 3 day period the best ACE-IMK coincidences are in a different latitude to the best ACE-ESA coincidences - I would expect them to be mostly the same locations. The averaging of all MIPAS profiles within the coincidence criteria also seems undesirable. The averaging will reduce the random noise in the MIPAS profiles so the contribution to the overall SD is no longer straightforward. Better to take just the closest profile. Also, noting the difference in time/space would allow subsequent analysis as to whether the chosen margins are adequate or, more ambitiously, allow the colocation error to be quantified e.g., when switching to grid boxes or zonal averages.

This was addressed in the response to reviewer G. Toon as follows. "In the revised version of the manuscript, when multiple MIPAS profiles are spatially coincident with an ACE-FTS profile, the MIPAS profile closest in time is selected. In addition, there must be both MIPAS IMK and MIPAS ESA processed data available for this coincident profile".

In lines 230-238 of the revised version of the paper, the new procedure to select coincident profiles is described. Please, see also lines 241-242 where we clarify that only points with triple observation are used.

2) Algorithm descriptions

In 2.1.1/2.1.2 the descriptions of the two retrievals seem to be taken directly from the source papers using their own terminology (possibly via the SPARC papers), with little effort to standardise the information let alone provide some interpretation in terms of possible impact on the comparisons.

For example, 'non-linear least-squares global-fitting technique with Tikhonov regularisation' (for IMK) and 'least-squares global fitting, using the Gauss-Newton approach modified with the Levenberg Marquardt method' plus 'a posteriori regularisation' (for ESA). The reader has

to work quite hard to understand whether or not these are essentially the same thing and hence whether significant differences may arise from these aspects of the algorithm.

Even on a more basic level:

- Does ESA-MIPAS retrieve $H_2O$ and HDO as log(VMR) or VMR (which affects how you should evalulate biases)?

- Does the IMK-MIPAS account for horizontal inhomogeneities in the line-of-sight direction and/or assume LTE?

- Do both use microwindows in the same spectral region?

Similarly with ACE-FTS, orbital altitude and inclination are measured but these are not given for Envisat. Spectral bands are given for ACE-FTS but not for MIPAS.

We will improve the descriptions of the algorithms and retrieval set-ups and make them more consistent to each other. In each case, we will also describe the changes applied since the previous data version.

Section 2 has been edited to add information about the algorithm descriptions of the two MIPAS and the ACE-FTS datasets.

3) Retrieval diagnostics
There is no reference here to the retrieval characteristics such as expected random noise, systematic errors or averaging kernels, at least typical values - these are not likely to change much except (for MIPAS) towards the poles where the atmospheric temperature is low.

The SD of the bias, for example, could be put in the context of the retrieval random errors, and the bias itself in terms of the overall systematic errors. Meanwhile the averaging kernel describes the ability of the retrieval to follow 'real' atmospheric variations, which has an impact on the correlations as well as the SD of the comparisons. The lack of an HDO time signature in the ESA data might be explained in terms of the averaging kernel.

We will add all available information to the revised version.

Section 2 has been edited to add information about the retrieval diagnostics of the two MIPAS and the ACE-FTS datasets.

4) CH4 consistency
Given the large discrepancy between ACE and the two MIPAS profiles in the stratosphere a simply self-consistency check would be to see how these compare with their equivalent CH4 retrievals, on the basis that ($H_2O + 2CH_4$) should be conserved.

While this is an interesting idea, it is beyond the scope of this paper and will be considered by the authors for future work.

No action performed.

5) The authors should be aware of the difference in LaTeX between a minus sign ($-$) and a dash (--) indicating a range of numbers. Where both positive and negative values are possible it would also help if '+' were added in front of the positive numbers to further distinguish them from the --. Thus, in L24: $-$8.6--+10.6 Incidentally, assuming this is taken from Table 2, the actual number in the table is -8.7. Further on this particular point, negative and positive biases don't have any particular meaning in this sentence, so I would just have said 'biases of up to 10%'.

Thank you very much for this suggestion. The changes that the referee proposes will be considered. By one hand and according to the AMT style, we will use the word "to" for indicating a range and en dashes (–) for numerical purposes. By the other hand, the biases will be referred in total percentage, when appropriate, along the text in the revised version of the manuscript.

The text has been revised and according to AMT style. The word "to" has been used for indicating the range and "en dashes" for negative values. See for example the table 1 in the line 401 of the revised version of the manuscript. We also have used "biased up to" for indicating biases without sign as for e.g. in lines 408, 411, 413.

6) There are numerous references to 'global' averages whereas I would suggest 'dataset' averages, or something similar, unless you really claim that your intercomparison dataset represents some sort of uniform sampling of the globe.

The term "global" will be clarified in the revised manuscript.

We concur with the reviewer. The text has been revised and the word "average" has been used instead of "global" several times as for example in lines 310, 315, 318, 320, 328, 334.

7) It is unclear whether or not the MIPAS-IMK product has been updated (these authors refer to 'V5H' and 'V5R' whereas Hogberg (2019) referred to V20 as being the newer version.

See our comment above. As a result of the analyses by Hoegberg et al., 2019 and Lossow et al., 2020, a new retrieval approach for HDO has been developed for MIPAS-IMK, which we present here. We will clarify which different versions have been used for the $H_2O$, HDO and derived delta-D results for all instruments.

Clarified in lines 91-93 and 109-114.

8) 'Standard Deviation' is already defined as the spread around some mean value, so 'debiased standard deviation' is just 'standard deviation' since we are already talking about mean differences (or 'the standard deviation of the relative bias' which is how it is described in the caption for Fig.4). Perhaps you thought SD might be confused with root-mean-square-difference? Also in Tables 1 and 2 this has become '1 \sigma Bias' which sounds like a different thing altogether, but I assume it's the same.

We will make sure to use the same technical terms throughout the paper. We have used the term "de-biased standard deviation" to make clear that it is the standard deviation around the mean difference (spread around the bias) and not the spread of the data sets themselves. What is called "1-sigma bias" in Tables 1 and 2 is meant to be the de-biased standard deviation. This will be updated, too.

The term "de-biased standard deviation" has been used along the paper instead of "1-sigma bias". See for example the table 1 (line 401) and the Fig. 4 caption (line 358).

It is important to mention that before in Fig 3 of the new version of the manuscript, we report the standard error of the mean (SEM) instead of the standard error as indicated in line 256-257. This is because the standard error gives an indication on the precision of the single measurements. In Fig. 3 we computed the averages of many profiles, due to the large number of data points, the comparisons are highly significant, and we are interested in their biases.

9) The time series plots could be enhanced by subtracting out the mean profile and then also removing the average annual cycle to show interannual variability. The latter may have some QBO correlation which could be investigated.

Again, we are intrigued by this suggestion for additional analyses. However, they are beyond the focus of this current paper.

No action performed.

10) Details of bias determination (3.1.3, 3.2) are, firstly, confusing because of the repeated use of the same 4 coordinates for each parameter, secondly difficult to read because of the small fonts and, finally, quite standard. Even then, there are a few problems here Eq 1 presents the bias as 4-coordinate average of b_i which are themselves 4-coordinate quantities. However, it seems unlikely that any two measurements are exactly matching in latitude or time (it's not clear what 'period' means here) so I assume these coordinates are what is being averaged over in i=1, n so should not appear on the l.h.s. as well. And b_i is presumably also a function of longitude, also averaged.

We agree with the referee in the confusing use of 4 coordinates for each parameter. In the revised version of the manuscript the notation is simplified using the formalism of Dupuy et al ACP, 9, 287–343 (2009).

Modified in lines 261 – 300.

sigma_x1 and sigma_x2 in Eq (7) are undefined,

Will be fixed.

Defined in line 291.

$\sigma_b$ has a bar over the b here but not in Fig 4.

Will be fixed.

Modified. Please see Fig. 4 in line 356.

Fig 4 shows SD as a percentage (of what?) but Eq 5 defines this in absolute terms.

The debiased standard deviation is calculated to the mean relative bias, therefore the unit is percentage. Eq (5) is defined in relative terms, it will be clarified in the revised version of the manuscript.

Clarified in lines 283-284 and 286.

When considering relative bias, if the two datasets were retrieved as log(VMR), the geometric rather than arithmetic mean of the two values would be more appropriate as the reference value.

Only one of the datasets is retrieved in log(VMR) so the arithmetic mean is applied in all cases.

Line 173 indicates that MIPAS-IMK data is retrieved in log(VMR). No more data is indicated to be retrieved in log(VMR).

I appreciate that this sort of thing was included in the previous papers (and had I reviewed those I would have said the same thing) but that's even less reason to include it again here. Everyone knows what you mean (and they can look up Pearson Correlation on Wikipedia) so no need to drag the reader through all the small print. On this (rare) occasion, it really is simpler to explain in words rather than equations.

For clarity on how the correlation calculation is performed, we have chosen to include this equation.

Now in line 290 of the revised version of the paper.

11) You could save a lot of wordage simply referring to these data as 'IMK', 'ESA' and 'ACE' (at least in the text, perhaps not in the figures)

Thank you for the idea. Since these abbreviations are not commonly used in the previous literature, we would prefer to maintain the references to the datasets as they are.

No action performed.

**Minor/Typographical comments**

Abstract

L139: Use a regular reference for the SPARC special issue to avoid the typesetting difficulties caused by putting the URL (http:...) in the text.

This is the standard method for referring to this special issue.

See line 85 of the revised version of the paper.

L185: 'from 1 to 70 km'

The referee is right! The word "since" will be changed by "from".

This information was duplicate, and this sentence has been deleted. It can be seen in line 246 of the revised version of the paper.

L189: Presumably the two MIPAS datasets are automatically colocated, so this just refers to colocating ACE-FTS with MIPAS.

See methodology response above.

Please, see our answer to the general comment #1 Colocations. In the revised version of the paper only points with triple observation are used as indicated in 241-242.

Fig 2: I assume the variation in low altitudes is due to cloud-screening but what causes the reduction in MIPAS-ESA comparison data at high altitude?

The reduced number of coincidences for the ESA profiles above 50 km is due to the fact that profiles from different observation modes are used, in particular measurements from the UTLS observation mode, which are about 8% of the total MIPAS observations and are performed mainly in the period August 2004/August 2005, are characterised by the altitude range 8.5 - 52 km.

However, as stated in the first comment, in the revised version of the manuscript we compare the same number of coincident profiles for the three databases. The number of coincidences in the stratosphere is 15263 and it decreases going to the lowest altitudes up to 4078 at 10 km.

See Fig 2 in line 246 of the revised version of the paper. The reduction from 40 km and upwards is because HDO ACE-FTS data decrease as made clear in line 250 of the revised version of the paper.

L201: It would be useful to have at least an approximate figure as to what percentage of MIPAS profiles fail the quality control tests.

For MIPAS-IMK about 0.16% of all started retrievals of HDO did not converge. About 0.05% of all started retrievals failed or encountered corrupted spectra. In total 2,314,011 profiles were processed. This means that 99.79% of all profiles are considered healthy.

The two flags that we provide with the data (visibility flag needs to be 1, and averaging kernel diagonal needs to be > 0.03) are meant to be applied to single points in the vertical profile, i.e., these flags reduce the altitude coverage of one profile, but leave the other parts of the profile valid. Please note that we always provide the profiles from 0 to 120 km. The flags define the valid altitude range.

MIPAS-ESA uses a different approach for filtering out bad profiles.

The quality of the retrieved profiles is judged "good" when three requirements are met: the retrieved profile adequately reproduces the measurements (i.e. the chi-square value at the final step of the iterative procedure is smaller than a pre-defined mode and species-dependent threshold), there are no outliers in the retrieval error (i.e. the maximum value of the retrieval error profile is smaller than a pre-defined mode and species-dependent threshold) and the iterative retrieval procedure successfully converges.

When at least one of the previous requirements is not verified, the whole retrieved profile is flagged as bad in the output file (post_quality_flag=1) and it is not used as either profile of an interfering species or initial guess in subsequent retrievals. Otherwise, if all previous conditions are verified, the post_quality_flag is set to 0 so that the retrieved profile is considered "good", and it can be used for subsequent retrievals. If the retrieved temperature is flagged as bad, no VMR retrieval is performed, since a proper temperature profile is fundamental for the retrieval of the trace species.

Each retrieved profile is properly and fully characterised on the full retrieval range provided in the output files by the corresponding CM and AKM. Altitude regions with poor information on the retrieved target can be identified by the low values of the diagonal elements of the AKM and/or the large values of the diagonal elements of the CM. Since the AKM and the CM are calculated considering the retrieval on the full vertical range, even if some of the retrieved values are discarded by the user, we recommend to use the full profile along with its full CM and AKM.

2.54 million HDO profiles are available in the products. The percentage of ESA good profiles is reported in Fig. 6 of Dinelli et al., 2021 paper for all retrieved trace species. 8% of HDO retrieval procedures fails.

Part of this information will be provided in the revised version of the paper.

Part of the above-mentioned information has been added in the section 2 of the revised version of the paper.

L208: In L187 the grid is from 1-70km. Assuming the IMK grid is at 1km intervals how do you get 58 levels?

The MIPAS-IMK grid is not strictly a 1-km grid. It is a 1-km grid from 0 to 44 km, followed by a 2-km step width from 46 to 70 km. These are 58 levels. We will correct this description in the revised version.

Description added in line 246-247 of the revised version of the paper.

L304: I would not refer to these as 'error bars', just 'bars'.

Will be changed. Thank you.

Done. See for example lines 311 and 335.

L319: Describing these values as 'global' minima is misleading, they're actually the minima in the intercomparison dataset.

The term "global" will be clarified in the revised manuscript.

These sentences about the minima in the intercomparison dataset have been removed as we have reconsidered that they are not very helpful.

L325: Similarly, 'global mean' profiles.

Also, here.

As suggested by the reviewer above, the term "average" is used instead of "global mean". See line 320 of the revised version of the paper.

Fig 3. These plots would be more informative (and take less space) if all three datasets were combined on the same plot, allowing MIPAS-IMK and MIPAS-ESA to be directly compared. Use eg dashed lines to mark the 1sigma variation for each (Also, I wouldn't call these 'error bars' as in the current caption).

We agree with the reviewer suggestion, the three curves will be combined in the same plot. We also improve the figure 3 including the MIPAS-IMK to MIPAS-ESA comparison. Regarding the "error bar", we think that the referee is right, they are "bars".

Figure 3 has been modified as well as its caption (see lines 333-335 of the revised version of the paper).

L365: It is perhaps worth mentioning that the low SD (Fig 4(c)) between ESA-ACE and coupled with the high correlation (Fig 4(d)) are consistent with the MIPAS-IMK retrievals being less sensitive to actual atmospheric variations in H2O, and conversely for IMK-ACE for HDO.

We agree with this statement for altitudes above ~25 and ~15 km for $H_2O$ and HDO, respectively, and will add it to the revised version.

Added in lines 348 – 351 and 367 - 371 of the revised version of the paper.

Fig 4(g) mis-labelled(?) as '\sigma_b Bias' (and similarly Table 1)

No, this notation is correct as it is the de-biased standard deviation.

Nevertheless, the notation has been changed to "sigma_b relative bias" as Eq 8 defines the de-biased standard deviation in relative terms (see Fig. 4 in line 358). The label of the table 1 was also changed (see line 401).

Table 1 & 2: I assume these figures are a summary of the plots shown in Fig 4, in which case make it explicit in the Table caption (and similarly in the Fig 4 caption refer to these tables). Rather than have rows labelled eg 'MIPAS-IMK vs 'ACE-FTS' I suggest replacing

'vs' with $-$ to make it absolutely clear which way around you are defining the sign of the bias.

These changes will be made in the revised paper.

The figure caption and the table caption have been modified (lines 358 and 401 respectively). In table 1, "vs" has been replaced by "–".

L393: Is there such a thing as a 1\sigma standard deviation? I though the SD was, by definition, 1 \sigma.

We agree with the referee. Our sentence is redundant. Will be changed.

This sentence has been removed in the revised version of the manuscript.

Fig 5 caption refers to 'meridional cross-sections' which suggests a single slice through a particular longitude. 'Zonal mean' or 'Zonally averaged' distributions are the usual terminology for such plots. And rather than use 'summer' and 'winter' - which differ from north to south hemispheres - give the actual months averaged (as in Fig 6).

These updates will be made.

"Latitude-altitude cross section" and "zonal mean" have been used in the revised version of the paper. See for example lines 430 (Fig. 5 caption) or 461.

The actual months averaged are used instead of "summer" and "winter" when possible, as in line 439, 445 or 456 of the revised version of the paper.

L405: I'm assuming that the labelling on Fig 5 is correct (I would have expected ACE-FTS to be missing the high latitude measurements during the local winter months but, on the other hand, they may also specifically target as high latitude as possible during polar winter months). However, the very low ACE values for both H2O and HDO in the winter polar vortex (compared to the two MIPAS datasets) are worthy of comment.

Yes, this figure is correct. ACE-FTS is in an orbit that targets high latitude measurements, more than 50% are at latitudes higher than 60 degrees, to investigate polar ozone chemistry. Note, that the local winter mean from ACE-FTS does not include data from all of these months because of the requirement for sunlight for its measurements. This requirement leads to the ACE values sampling only the later part of this season (in austral winter, mainly Aug. in JJA at the highest latitudes) than the two MIPAS datasets sampling all months. It is likely this sampling difference that leads to ACE-FTS showing more dehydration than MIPAS in these zonal mean plots.

Commented in lines 438-444 of the revised version of the manuscript.

Also, why is the IMK cloud filtering any different for HDO than for $H_2O$?

Lossow et al., 2020 have made sensitivity tests regarding the retrieval of HDO from MIPAS data. They found that for the HDO retrieval, the upward error propagation was pronounced at the lower end of the profiles in previous data versions. I.e., incorrectly retrieved vmrs due to, e.g. unidentified cloud contamination trigger retrieval errors in the vmrs above this altitude. To be on the safe side, Lossow et al., 2020 recommended to discard the retrieved values in the altitude range of the lowest two (V5H) to three (V5R) tangent altitudes. They found that the propagated error fades out sufficiently above this level so that data from levels above can be used.

In the caption of figure 5 we have included the sentence "the absence of profiles in MIPAS-IMK map below the tropical tropopause is due to a more stringent cloud filtering approach used by IMK".

L450: 'interannual variability' implies what's left after removing the annual cycle. From Figs 7a,b,c alone it is not clear to me that ACE-IMK have the greatest similarity.

The referee is right, we are not analyzing the "interannual variability" (seasonal cycle subtracted from the data sets) but the time series themselves. This sentence will be changed.

"Interannual variability" is removed. In lines 471-472 of the revised version of the paper, we use "annual variation".

The referee is also right again, while the tape-recording effect is clearly seen in the MIPAS-IMK HDO time series, this is less evident in both MIPAS-ESA and ACE time series.

Modified in lines 468-472 ($H_2O$), 479-482 (HDO) and lines 486-488 ($\delta D$) of the revised version of the manuscript.

L465: 'shows' rather than 'shown'.

Will be changed.

In the new version of the manuscript, we restrict the climatological analysis to the range between 16 km and 30 km (the lower and middle stratosphere). Please see lines 331-332 Therefore, this sentence about bias at the troposphere in the altitude-time diagrams has been removed.

L467: 'while the ACE-FTS instrument can measure with higher sensitivity ...' What is the basis for this statement?

ACE-FTS has a higher sensitivity due to the combination of the long-path length through the atmosphere in limb-view and the solar occultation measurement technique used. This makes the ACE-FTS measurements less susceptible to perturbations due to thin cirrus clouds in the UTLS.

This sentence in the old line 467 has been removed for the same reason that in the comment above.

L485: 'vertical behavior' (or 'behaviour') presumably just refers to the small mean profile bias averaged over the whole dataset, shown by the red-lines in Fig 4a,b. 'Behaviour' suggests that the two track each other well in other respects such as low SD and high correlation, which they don't (at least not above 20km).

This sentence will be clarified in the revised manuscript.

"The vertical behavior of $H_2O$ profiles between 16 and 30 km altitude levels shows…" has been replaced by "The mean profiles of $H_2O$, HDO, and δD between 16 and 30 km, averaged over all latitudes, show…" (line 514 of the revised version of the manuscript).

L487: 'according to the uncertainties' Which uncertainties are these? There was no reference to estimated uncertainties in the datasets in the main part of the paper.

Given that the two MIPAS profiles are 1ppmv higher than ACE after averaging over several thousand profiles suggests rather a significant discrepancy to me.

As mentioned above, we will provide all available information on uncertainties in the revised paper.

The sentence of the old line 487 has been reworded without using "uncertainties" (lines 515-517).

L495: 'quantitaty'?

Will be fixed.

Modified in line 494 of the revised version of the manuscript.

L525: 'MIPAS-IMK dataset provides a more realistic signal for the entire stratosphere' This is a bold statement and needs some qualification otherwise it could get quoted out of context. Is this for H2O, HDO, delta D or all three? Is it in terms of the time-evolution or mean profile? Is it because it has the best correlation or lowest SD compared to both other datasets?

"More realistic than MIPAS-ESA" was to be meant and below the 30 km of altitude. However, we agree with the reviewer and even with all the context the statement can be a bold given the existing data. We reword this paragraph in the revised version of the manuscript.

This paragraph has been removed in the revised version of the paper.

---

## Referee Report (RR1)

Referee comments Feb 2024

Comparison of the H2O, HDO and delta D stratospheric climatologies between
the MIPAS-ESA v8, MIPAS-IMK v5 and ACE-FTS v4.1/4.2 satellite data sets

by K de los Rios et al.

Overview

The paper compares three different datasets for (primarily) stratospheric
H2O, HDO and the derived delta D for the period 2004-2012 using latest
processed version of data from the IMK and ESA retrievals of MIPAS data,
and the ACE-FTS. This builds on previous work by other authors who used
older and more limited datasets. In particular the extended time period
illustrates how the H2O 'tape recorder' effect in the equatorial stratosphere
is represented with very different levels of clarity.

Main Comments

Overall this seems to be more of a technical report rather than a scientific
paper: the data are read, the recommended screening is applied, the
results are plotted, analysed and discussed. There's nothing wrong with it,
as such, but the authors miss some opportunities for providing new insights.

I list a number of suggestions, which the authors may wish to consider,
which I think would improve the paper.

1) The descriptions of the algorithms and retrieval characteristics
behind the different datasets read very much like extracts from the
separate source papers, including many obscure technical details.  I
prefer to have seen a single, shorter and, most importantly, original
description highlighting the similarities and differences where they
might be relevant to the results presented, which would also show that
the authors have applied some critical understanding of the technical
details rather than simply relaying the information to the reader to
evaluate.

2) The averaging kernels, in particular, seem key. There really should
be a figure allowing these to be compared rather than verbal
descriptions of the two MIPAS AKs and nothing at all regarding
ACE-FTS.  I couldn't find any mention of whether the ACE-FTS
retrievals use any kind of regularisation and/or climatological a
priori constraint, and I would expect the authors to have at least
asked themselves the same question.

And, from the AKs of H2O and HDO, one point that could have been developed
is how to determine the AK for delta D.

3) Given that MIPAS-ESA and MIPAS-IMK both use fundamentally the same
set of observations, the comparisons would have been simpler if MIPAS
profiles were *only* used when data from both processors were
avaiable. There would be some loss of data from the UTLS-1 and AE
modes, but negligible compared with the advantage of eliminating sampling bias.

4) While the systematic errors for both the MIPAS-IMK and MIPAS-ESA
retrievals are dominated by spectroscropic uncertainties, and it is
established that there are some difference in the H2O spectroscopic
data used, it seems most unlikely that the spectroscopic data are so
independent that they account for much of the difference between these
two. A plot of the H2O and differences using the two spectroscopic
databases, with microwindows marked, say for 20km altitude would have
helped answer this.

5) Another point that wasn't addressed was whether there was any significant

difference between the day and night profiles for the two MIPAS datasets.
$H_2O$, in particular, has a strong non-LTE signature in the stratosphere
and this could lead to spurious day-night differences in the results
(with, presumably, the night profiles being less affected).
This could simply be incorporated into Fig 3 and may explain some of the
difference between the MIPAS-ESA and MIPAS-IMK processors.

6) For the debiased SD (Fig 4) this should show some correspondence with the
sum of the random errors associated with the individual profiles, ie
sqrt( e_1^2 + e_2^2 ) where e_1 and e_2 are the reported random errors.
It would be useful to have these plotted on the same figure for comparison.
Another diagnostic would be to show the actual SD of each dataset about
its mean. A certain amount of this would be atmospheric variability - presumably
the same for all three instruments, but subtracting some variability due to the
regularisation while adding variation due to the instrument random noise.
One could go further: if it is assumed that the bias is constant, three sets
of comparisons between three datasets is enough information to assign as SD
to each dataset. Thus from the debiased SD results, one can empirically
determine the actual SD of each dataset about its mean bias.

7) Sections 3.1.1 and 3.2 should be an appendix. While useful for the
purpose of defining terms, this is just standard statistics.

Minor Comments

L34: This is not a particularly controversial or original statement, so I
   would suggest "(e.g., Hegglin et al ... "

L42: This statement probably does need a supporting reference, instead of
    just 'Scientists discovered it'. Rosenlof et al will probably suffice
    but use that at the start of the paragraph.

L23: lower biases for HDO, but not delta D. But you should also mention that
    the smallest bias for $H_2O$ is between MIPAS-IMK and MIPAS-ESA.

L24: I would interpret a 'meridional cross-sections' to be cross-sections
    at a particular longitude, but here you really mean 'zonally averaged
    cross-sections'

L61: "Atmospheric limb-sounding" would be better than "Limb Earth probing".

L71: "highly reliable" is unecessary and rather subjective. Perhaps just say
    "made regular WV observations..." (and use past tense).

L75: Since you have already described Odin, I think it would be useful at
    this point to briefly mention that MIPAS made continuous observations
    of the infrared limb-emission, obtaining around 1000 profiles a day with
    global coverage, while ACE-FTS used solar occultation which gave typically
    28 profiles a day split into two narrow latitude bands (which varied
    throughout the year).

L83: Suggest "e.g.," or "i.e.," instead of "like".

L85: references to web-pages should probably appear as usual citations rather
   than directly within the body of the text (unless AMT has its own rule on
   this). Also L156, L174,

L85: Suggest 'latest' rather than 'last' - they may want to produce another.

L85-L102: There is a rather confusing mass of detail over specific datasets
    here, much of which is repeated in Section 2. For this part, the
    introduction, the emphasis should be on clarity so try to remove some
    of the obscuring details which are covered in Section 2.
    (Even in section 2 I feel it would be more clearly represented in a table

listing dataset, date range, products compared and the reference).

L113: v4.2?

L136: What is the MIPAS FOV width? (and that of ACE-FTS?)

L148: A table listing the microwindows would have be useful

L149: I don't know why information on molecules other than H2O is provided here
  - are they expected to have a significant contribution to the results?
  OCS, for example, only has a significant absorption feature around 2100cm-1,
  well outside any of the spectral regions used for the H2O retrieval?

L154: Since these links refer to images you should include them directly in
  the manuscript (or else replot the data), otherwise this paper will be
  incomplete if the links ever disappear.

L157: 'is about 3 km'

L236: Since both MIPAS processors have used the same set of spectra, the
  differences in time/location are purely due to how these values are assigned
  to the resulting L2 profiles.

Fig1: Why would the MIPAS-ESA and MIPAS-IMK profile locations be any different?

L246: This is inconsistent. Is the grid from 0-70km or 1-70km? Is it 1km spacing
  up to 44km or up to 46km?

L260: Another approach you could have considered is averaging ln(H2O)
  and ln(HDO) (assuming the values are always constrained to be
  positive).  Since there is a strong variation with height in the
  tropopause this avoids biasing towards large values in the
  average. This may explain some of the behaviour of the MIPAS-ESA HDO
  profile at low altitude in Fig 3.

L326: "along the stratosphere" - what does this mean? Along usually indicates a
   horizontal direction.

Fig 3: With >1000 profiles compared over most of the altitude range I think we
  can assume that the standard error will be negligible, so the error bars
  just clutter the plot.

Table 1: this would be clearer if the columns were lined up, eg split each into
   two columns, min and max, and use + signs for positive values. Also I don't
   think more than 1 significant figure is justified, certainly not 4 as used
   for the absolute bias of delta D.

L406-415: Table 1 already summarises the previous plots so I don't think yet
  more text summarising Table 1 is required.

Fig 5: "during I boreal"

Fig 5: "The climatology is based ..." - presumably you are referring to these
  plots as "the climatology" but the plots are introduced as "latitude-altitude
  cross-sections" not as a "climatology". Perhaps if you write "This
  climatology  is based..." it establishes what you meant.

Fig 6: These plots might be clearer if presented as deviations from the mean
  profile. It's hard to distinguish the various shades of blue/green which
  contain the signal for H2O and HDO.

Typographcial inconsistencies
 - Both upper and lower case for version, eg v8 in title, V8 in abstract

- Water vapor (eg L15) and water vapour (eg L16)

  - Data set (eg title) and dataset (eg L69)

  - Use '--' in LaTeX to indicate a range of numbers, not hyphens (eg L157).

---

## Referee Report (RR2)

| Instrument | Dataset | Dates | Altitude | Coverage | Spectroscopy |
|---|---|---|---|---|---|
| | | | km | cm$^{-1}$ | |
| MIPAS-IMK | V5H_H2O_20
V5R_H2O_220/221
V5H_HDO_22
V5R_HDO_222/223 | 2002-2004
2005-2012
2002-2004
2005-2012 | 5-72 | $H_2O$: 795-827
& 1223-1410
HDO:1250-1482 | MIPAS_pf3.32
(HITRAN 1996) |
| MIPAS-ESA | L1 V8.22 | 200?-???? | 5-55 | $H_2O$: 783-956
& 1224-1696
HDO: 1218-1471 | HITRAN_mipas_pf4.45
(HITRAN 08/12/16) |
| ACE-FTS | V4.1/4.2 | 2004-present | 5-150 | $H_2O$: 937-3173
HDO: 1383-1511
& 2493-2673 | HITRAN 2016 (?) |

Table X. Key aspects of the three retrieval methodologies compared in this work.

---

## Author Response (AR2)

**Revision of "Comparison of the H2O, HDO and δD stratospheric climatologies between the MIPAS-ESA v8, MIPAS-IMK v5 and ACE-FTS V4.1/4.2 satellite data sets" by De los Rios et al., submitted to AMT.**

Dear Editor,

In this revision the table suggested by the reviewers is included. The text has been shortened from 561 lines without the table to 539 lines including the added table. All the comments by both reviewers have been considered and replied in red colour.

With best regards, on behalf of all authors,
Paulina Ordoñez

**Reviewer #1**

I agree that significant improvements have been made to the manuscript, to the point that it is now publishable in my opinion. That said, the manuscript is still unnecessarily long and parts are difficult to follow. In the technical comments below, I have made suggestions how to slightly shorten the paper.

In order to better follow the discussion of the datasets, I made a table for myself summarizing the key information. I found myself continually referring to this table, so perhaps it (or something similar) should be part of the paper itself. Readers not familiar with MIPAS processing might benefit.

Thank you very much for the table, this is very useful. The reviewer #2 also comments that we should summarize this information. We include the following table in the revised paper.

| Instrument | Molecule | Dataset | Dates | Altitude (km) | Coverage (cm$^{-1}$) | Spectroscopic database | Microwindows |
|---|---|---|---|---|---|---|---|
| **MIPAS-IMK** | **H$_2$O** | V5H_H$_2$O_20 V5R_H$_2$O_220/221 | 2002-2004 2005-2012 | 5-72 | 795-827 1223-1410 | MIPAS_pf3.32 (HITRAN 1996) | Von Clarmann et al. (2009). |
| | **HDO** | V5H_HDO_22 V5R_H$_2$O_222/223 | 2002-2004 2005-2012 | 5-72 | 1250-1482 | | Steinwagner et al. (2007) |
| **MIPAS-ESA** | **H$_2$O** | MIPAS L2V8 | 2002-2012 | 5-55 | 783-956 1224-1696 | HITRAN_mipas_ pf4.45 (HITRAN 2012) | Dinelli et al (2021) |
| | **HDO** | MIPAS L2V8 | 2002-2012 | 5-55 | 1218-1471 | | Dinelli et al (2021) |
| **ACE-FTS** | **H$_2$O** | V 4.1/4.2 | 2004-present | 5-150 | 937-945 1195-1990 3151-3173 | HITRAN 2016 | Boone et al. (2017) |
| | **HDO** | V 4.1/4.2 | 2004-present | 5-42/50* | 1383-1511 2605-2673 | HITRAN 2016 | Boone et al. (2017) |

Key aspects of the three datasets compared in this work.
* Upper altitude of retrieved profile differs between polar (42 km) and equatorial (50 km) latitudes.

Technical comments.

Line 23: "For HDO and δD, lower biases are found in the MIPAS-ESA and ACE-FTS comparison". When I look at Fig.3b, MIPAS-ESA HDO is an outlier above 35 km. Perhaps the authors mean that lower biases are found in the MIPAS-IMK and ACE-FTS comparison?

We were referring to the bias analysis results in line 23 (Fig 4), but also it is below 30 km. We have clarified it in lines 22-24 of the revised version of the manuscript.

Line 48:" ...accompanied by large horizontal motions to mid-stratospheric latitudes ". I don't understand. Perhaps the authors mean "accompanied by large horizontal motions at mid-stratospheric latitudes".

The reviewer is right, and the sentence is now changed to "accompanied by large horizontal transport at mid-stratospheric latitudes".

Line 127: The "NOM", "UTLS-1", and "Aircraft Emissions" observation modes are introduced here along with their altitude ranges. "NOM" is used only once more. "UTLS-1" and "Aircraft Emissions" modes are never used again. So, I suggest stating that the nominal observational mode, covering 5-72 km, is used in this work. No need to tell us about the other modes.

MIPAS-IMK data are for NOM mode only which was mentioned in line 178-179 of the manuscript.
MIPAS-ESA uses all these modes as mentioned in line 145 "HDO has been retrieved from all the observation modes listed above".

Line 146: "the ones" --> "those"

Done.

Line 147: "lies" --> "lie"

Done.

Line 157: " H2O vertical resolution is about 3 at 10 km, then it slowly degrades, reaching 5 - 6 km at 20 km, 7.5 at 30 - 40 km, 10 at 50 km." These are 3 instances of missing "km" and a missing "and" in this sentence. It should be " H2O vertical resolution is about 3 km at 10 km, then it slowly degrades, reaching 5-6 km at 20 km, 7.5 km at 30 - 40 km, and 10 km at 50 km."

Done.

Line 162: "The relative average single scan random error varies with altitude for the different atmospheres, but...". Which different atmospheres?

Thank you for the comment. We were meaning "for different atmospheric conditions". It has been modified in line 157 of the revised version of the paper.

Line 173: "MIPAS-IMK WV retrievals used here were retrieved in log (VMR) space" But line 177 states that "HDO was retrieved in linear space". So, is HDO not WV?

MIPAS-IMK retrieves the main isotopologue of water vapor on log (VMR) space (it is clarified in line 167 of the revised version of the manuscript), but not HDO.

Line 180: "omitted" --> 'avoided".

Done

Line 181: "the vertical resolution of δD is provided by the difference between the a priori and the retrieved profile". I don't understand this at all.

As a characteristic of the Tikhonov regularisation that smoothes the retrieved profiles only, the structures in the a priori profile provided by the main isotopologue retrieval are smoothed in the HDO retrieval according to its vertical resolution (Speidel et al., 2018). We clarified it in line 174-176 of the revised version of the manuscript.

Line 185: "range." --> "ranges."

Done

Lines 186-189: "data base" --> "database" (3 instances).

Thank you. We use always "database" in the revised version of the paper.

Line 215: "uses a minimum altitude spacing of 2 km for tangent heights above 15 km and a minimum spacing of 1 km for tangent heights below 15 km." --> "uses minimum altitude

spacings of 2 km for tangent heights above 15 km and 1 km for tangent heights below 15 km."

Done.

Line 231: "For the coincidence pairs, the ACE-FTS data, which is the sparser dataset in the tropics was used as the first data set." I don't understand why the order matters.

It is a matter of efficiency. If we start with MIPAS profiles and search for an ACE-FTS profile for each of the millions of MIPAS profiles, many fails will probably be gotten. If we search MIPAS profiles for each of the several ten thousand of ACE profiles, a MIPAS profile for almost all ACE-profiles will probably be gotten. In the first case, the loop goes over millions of cases, in the second case it goes over ten thousand. However, the reviewer is right, and the result should indeed be the same. Therefore, this sentence is omitted in the revised version of the manuscript.

Line 250: "The number of ACE-FTS HDO profiles decreases from 40 km of altitude and upwards." --> "The number of ACE-FTS HDO profiles decreases above 40 km altitude."

Done.

Line 254: I don't understand why 3 colors are needed. Or why the y-scale extends to 60 km when the largest y-value is only 48 km.

One colour is now used in Fig 2 and the y-scale extends to 50 km.

Line 257: "sample" --> "sample size"

Done.

Line 390: Add "," after "ACE_FTS".

Done.

Lines 494 to 501: This paragraph seems to repeat parts of the introduction. It does not relate to the work that you did. I suggest deleting.

Deleted.

Line 517: "upwards 30 km of altitude" --> "above 30 km altitude"

Done.

Line 530: "9 years" --> "8 years"

Done.

Line 554: Add "," after "δD".

Done.

Line 559: "on a long period" --> "over a long period"

Done.

**Reviewer #2**

Comparison of the H2O, HDO and delta D stratospheric climatologies between the MIPAS-ESA v8, MIPAS-IMK v5 and ACE-FTS v4.1/4.2 satellite data sets by K de los Rios et al.

Overview

The paper compares three different datasets for (primarily) stratospheric H2O, HDO and the derived delta D for the period 2004-2012 using latest processed version of data from the IMK and ESA retrievals of MIPAS data, and the ACE-FTS. This builds on previous work by other authors who used older and more limited datasets. In particular the extended time period illustrates how the H2O 'tape recorder' effect in the equatorial stratosphere is represented with very different levels of clarity.

Main Comments

Overall this seems to be more of a technical report rather than a scientific paper: the data are read, the recommended screening is applied, the results are plotted, analysed and discussed. There's nothing wrong with it, as such, but the authors miss some opportunities for providing new insights.

I list a number of suggestions, which the authors may wish to consider, which I think would improve the paper.

1) The descriptions of the algorithms and retrieval characteristics behind the different datasets read very much like extracts from the separate source papers, including many obscure technical details. I prefer to have seen a single, shorter and, most importantly, original description highlighting the similarities and differences where they might be relevant to the results presented, which would also show that the authors have applied some critical understanding of the technical details rather than simply relaying the information to the reader to evaluate.

We have shortened the text and added the following summarizing table with the characteristics of the algorithm and the retrieval diagnostics, as also suggested by the reviewer G. Toon on his comments. In this way the similarities and differences are clearer.

| Instrument | Molecule | Dataset | Dates | Altitude (km) | Coverage (cm$^{-1}$) | Spectroscopic database | Microwindows |
|---|---|---|---|---|---|---|---|
| MIPAS-IMK | H$_2$O | V5H_H$_2$O_20 V5R_H$_2$O_220/221 | 2002-2004 2005-2012 | 5-72 | 795-827 1223-1410 | MIPAS_pf3.32 (HITRAN 1996) | Von Clarmann et al. (2009). |
| | HDO | V5H_HDO_22 V5R_H$_2$O_222/223 | 2002-2004 2005-2012 | 5-72 | 1250-1482 | | Steinwagner et al. (2007) |
| MIPAS-ESA | H$_2$O | MIPAS L2V8 | 2002-2012 | 5-55 | 783-956 1224-1696 | HITRAN_mipas_pf4.45 (HITRAN 2012) | Dinelli et al (2021) |
| | HDO | MIPAS L2V8 | 2002-2012 | 5-55 | 1218-1471 | | Dinelli et al (2021) |
| ACE-FTS | H$_2$O | V 4.1/4.2 | 2004-present | 5-150 | 937-945 1195-1990 3151-3173 | HITRAN 2016 | Boone et al. (2017) |
| | HDO | V 4.1/4.2 | 2004-present | 5-42/50* | 1383-1511 2605-2673 | HITRAN 2016 | Boone et al. (2017) |

Key aspects of the three datasets compared in this work.
* Upper altitude of retrieved profile differs between polar (42 km) and equatorial (50 km) latitudes.

2) The averaging kernels, in particular, seem key. There really should be a figure allowing these to be compared rather than verbal descriptions of the two MIPAS AKs and nothing at all regarding ACE-FTS. I couldn't find any mention of whether the ACE-FTS retrievals use any kind of regularisation and/or climatological a priori constraint, and I would expect the authors to have at least asked themselves the same question.

ACE-FTS does not provide averaging kernels and it does not use any regularization or a priori constraint. For this reason, we do not provide any figure with AKs comparisons. The following has been added to the text: "Unlike the MIPAS-IMK and MIPAS-ESA retrievals, the ACE-FTS retrieval does not use any regularization (lines 202-203)".

And, from the AKs of H2O and HDO, one point that could have been developed is how to determine the AK for delta D.

As commented above, ACE-FTS does not provide AKs. For both MIPAS datasets this is an open question that requires further investigations.

3) Given that MIPAS-ESA and MIPAS-IMK both use fundamentally the same set of observations, the comparisons would have been simpler if MIPAS profiles were *only* used when data from both processors were available. There would be some loss of data from the UTLS-1 and AE modes, but negligible compared with the advantage of eliminating sampling bias.

We performed different tests before deciding to use all the data in figures 5 and 6.

In the figure 1 the vertical propagation of the tropical signal along the monthly evolution of the MIPAS-IMK and MIPAS-ESA data is shown. In this plot MIPAS profiles were used only when data from both processors were available. It can be seen that the differences with the plots included in the paper are small, and the conclusions that can be obtained are quite similar. However, the data gaps as in MIPAS-IMK also showed up in MIPAS-ESA.

[Figure]

**Figure 1.** Altitude vs. time diagrams over 30 S and 30 N of $H_2O$, HDO and δD for the datasets MIPAS-IMK, and MIPAS-ESA.

Plots were also performed only with coincident profiles from the three datasets as depicted in figures 2 and 3. The results are very noisy particularly in the case of the temporal evolution plots.

[Figure]

**Figure 2.** Latitude-altitude cross sections of H₂O in (a) boreal summer (JJA) and (b) boreal winter (DJF), HDO for (c) boreal summer and (d) boreal winter and δD during I boreal summer and (f) boreal winter for the three datasets.

[Figure]

**Figure 3** Altitude vs. time diagrams over 30 S and 30 N of (ACE-FTS), (b) MIPAS-IMK and (c) MIPAS-ESA datasets for H₂O (left column), HDO (middle column) and δD (right column).

4) While the systematic errors for both the MIPAS-IMK and MIPAS-ESA retrievals are dominated by spectroscropic uncertainties, and it is established that there are some differences in the H2O spectroscopic data used, it seems most unlikely that the spectroscopic data are so independent that they account for much of the difference between these two. A plot of the H2O and differences using the two spectroscopic databases, with microwindows marked, say for 20 km altitude would have helped answer this.

The spectroscopic databases used in retrievals are reported for each dataset, for example for MIPAS-IMK, this is von Clarmann et al. (2009) for $H_2O$ and Steinwagner et al. (2007) for HDO. A further investigation is beyond the scope of this paper.

Nevertheless, below you can find an example of differences in the retrieved $H_2O$ profiles when using different spectroscopic databases. The impact of the spectroscopic database seems quite large.

**Profile differences due to changed spectroscopic line database**

Retrieved profile differences are relevant only for H2O VMR. Differences in p,T and other species are much smaller (more than one order of magnitude) than the noise error. Average (solid blue) and standard deviation (dashed blue) of the H2O VMR differences are shown in the plot below along with the noise error predicted by the VCM calculated by the ORM (solid red).

[Figure]

5) Another point that wasn't addressed was whether there was any significant difference between the day and night profiles for the two MIPAS datasets. H2O, in particular, has a strong non-LTE signature in the stratosphere and this could lead to spurious day-night differences in the results (with, presumably, the night profiles being less affected). This could simply be incorporated into Fig 3 and may explain some of the difference between the MIPAS-ESA and MIPAS-IMK processors.

We know that there are non-LTE effects in the data above 40 km. Please see Stiller et al (2012) and Nedoluha et al (2017) where this has been discussed. As we should shorten the text, we prefer not to include a discussion of non-LTE effects in this paper.

- Stiller et al., 2012. Validation of MIPAS IMK/IAA temperature, water vapor, and ozone profiles with MOHAVE-2009 campaign measurements. https://doi.org/10.5194/amt-5-289-2012

- Nedoluha et al., 2017. The SPARC water vapor assessment II: intercomparison of satellite and ground-based microwave measurements. https://doi.org/10.5194/acp-17-14543-2017

6) For the debiased SD (Fig 4) this should show some correspondence with the sum of the random errors associated with the individual profiles, ie sqrt( e_1^2 + e_2^2 ) where e_1 and e_2 are the reported random errors. It would be useful to have these plotted on the same figure for comparison.

The SD of the bias has been put in the context of the retrieval random errors, as it is mentioned in lines 343 to 348 in case of Fig.4c; and in lines 361 to 368 in Fig. 4g.

Another diagnostic would be to show the actual SD of each dataset about its mean. A certain amount of this would be atmospheric variability – presumably the same for all three instruments, but subtracting some variability due to the regularisation while adding variation due to the instrument random noise. One could go further: if it is assumed that the bias is constant, three sets of comparisons between three datasets is enough information to assign as SD to each dataset. Thus, from the debiased SD results, one can empirically determine the actual SD of each dataset about its mean bias.

This is completely beyond reach for this paper.

7) Sections 3.1.1 and 3.2 should be an appendix. While useful for the purpose of defining terms, this is just standard statistics.

We respectfully disagree. We prefer to include these details in the paper so that everyone is clear on their use. Therefore, we prefer to maintain sections 3.1.1 and 3.2 in the main text.

Minor Comments

L34: This is not a particularly controversial or original statement, so I would suggest "(e.g., Hegglin et al ... "

Done.

L42: This statement probably does need a supporting reference, instead of just 'Scientists discovered it'. Rosenlof et al will probably suffice but use that at the start of the paragraph.

This sentence has been rewritten. "At the beginning of the century, an increase of the water vapour in the lower stratosphere in the last decades was proven. However, the reason for this humidification was not understood (Rosenlof et al., 2001 and references therein)". (lines 42-43).

Rosenlof et al (2001) contains several contemporary references. For example, they cited that "Oltmans et al. (2000) shows an increase in stratospheric water vapour at Boulder CO (40°N) of -1%/yr (0.05 ppmv/yr) over a 20-yr period using the frost point hygrometer of the NOAA Climate Monitoring and Diagnostic Laboratory (CMDL)", "Water vapor increase have also been documented from the Atmospheric Trace Molecule Spectroscopic (ATMOS) instrument [Michelsen et al., 2000], from combined multiple in situ measurement [Engel et al., 1996], and from the Halogen Occultation Experiment (HALOE) [Nedoluheat al.,1998; Smith et al., 2000]".
Rosenlof et al (2001), combined ten stratospheric water vapor datasets (including WV from older in situ data) to show that increases in stratospheric WV had persisted since the mid-1950s.

L23: lower biases for HDO, but not delta D. But you should also mention that the smallest bias for H2O is between MIPAS-IMK and MIPAS-ESA.

The smallest bias for $H_2O$ is between MIPAS-IMK and ACE-FTS for the range 16 km - 30 km, and it is mentioned in the lines 23-24 of the revised version of the paper.

L24: I would interpret a 'meridional cross-sections' to be cross-sections at a particular longitude, but here you really mean 'zonally averaged cross-sections'

Modified.

L61: "Atmospheric limb-sounding" would be better than "Limb Earth probing".

Modified.

L71: "highly reliable" is unnecessary and rather subjective. Perhaps just say "made regular WV observations..." (and use past tense).

We provided references to say that MIPAS WV observations are highly reliable. Anyway, in the revised version of the manuscript "made regular WV observations" is said.

L75: Since you have already described Odin, I think it would be useful at this point to briefly mention that MIPAS made continuous observations of the infrared limb-emission, obtaining around 1000 profiles a day with global coverage, while ACE-FTS used solar occultation which gave typically 28 profiles a day split into two narrow latitude bands (which varied throughout the year).

Odin is mentioned in the introduction because this instrument measures $H_2O$ and HDO. However, as their measurements are not simultaneous, δD can't be derived and these data are not used in this work.
Regarding the number of observations per day, this information was in section 2 for ACE-FTS (now in lines 192-193) and it is added for MIPAS in line 120-121 of section 2.

L83: Suggest "e.g.," or "i.e.," instead of "like".

Done.

L85: references to web-pages should probably appear as usual citations rather than directly within the body of the text (unless AMT has its own rule on this). Also L156, L174.

As we clarified in the previous revision, this is the standard method for referring to WCRP/SPARC II special issue (also valid for line 174).

Citation in line 156 is modified.
"https://earth.esa.int/eogateway/documents/20142/37627/README_V8_issue_1.1_20210916.pdf" is changed by "Raspollini et al (2020)".

- Raspollini, P., A. Piro, D. Hubert, A. Keppens, J.-C. Lambert, G. Wetzel, D. Moore, S. Ceccherini, M. Gai, F. Barbara, N. Zoppetti, with MIPAS Quality Working Group, MIPAS validation teams, MIPAS IDEAS+ (Instrument Data quality Evaluation and Analysis Service) team. ENVIromental SATellite (ENVISAT) MICHELSON INTERFEROMETER for PASSIVE ATMOSPHERIC SOUNDING (MIPAS). ESA Level 2 version 8.22 products - Product Quality Readme File. ESA-EOPG-EBA-TN-5, issue 1.0 [online]. Available from:
  https://earth.esa.int/eogateway/documents/20142/37627/README_V8_issue_1.1_20210916.pdf, accessed 29 February 2024, 2020.

L85: Suggest 'latest' rather than 'last' - they may want to produce another.

Changed.

L85-L102: There is a rather confusing mass of detail over specific datasets here, much of which is repeated in Section 2. For this part, the introduction, the emphasis should be on clarity so try to remove some of the obscuring details which are covered in Section 2. (Even in section 2 I feel it would be more clearly represented in a table listing dataset, date range, products compared and the reference).

The paragraph has been modified in the revised version of the manuscript, trying to avoid some details that are covered in section 2. By the other hand, please see reviewer#2 comment#1 for the summarizing table that are represented in the revised version of the manuscript.

L113: v4.2?

Yes, thank you. Modified.

L136: What is the MIPAS FOV width? (and that of ACE-FTS?)

It is 3 km in the vertical and 30 km in the horizontal for MIPAS and this is 3 km circular FOV at the limb for ACE, as mentioned now in lines 136 and 195 respectively.

L148: A table listing the microwindows would have be useful.

We have also added the original publications references for microwindows in the above depicted table. In our opinion it is not necessary to repeat all this information.

L149: I don't know why information on molecules other than H2O is provided here - are they expected to have a significant contribution to the results? OCS, for example, only has a significant absorption feature around 2100cm-1, well outside any of the spectral regions used for the H2O retrieval?

In the revised version of the paper, only the information about $H_2O$ is included in this paragraph.

L154: Since these links refer to images you should include them directly in the manuscript (or else replot the data), otherwise this paper will be incomplete if the links ever disappear.

The links to refer images has been changed by this cite:

Anu Dudhia, MIPAS Level 2 error analysis [online]. Available from: http://eodg.atm.ox.ac.uk/MIPAS/err/, accessed 29 February 2024, 2020.

L157: 'is about 3 km'

Thank you, "km" is added.

L236: Since both MIPAS processors have used the same set of spectra, the differences in time/location are purely due to how these values are assigned to the resulting L2 profiles.

Different L1b versions have slightly different geolocations and times, and therefore we have applied this coincidence criteria to make sure we collect the same profiles from the two data sets.
No action performed.

Fig1: Why would the MIPAS-ESA and MIPAS-IMK profile locations be any different?

We use the same colour for MIPAS in the Fig 1 of the revised version of the paper.

L246: This is inconsistent. Is the grid from 0-70km or 1-70km? Is it 1km spacing up to 44km or up to 46km?

The reviewer is right. The information was inconsistent because it was wrong.
It's formally from 0 to 70 km, however, we don't have a measurement at the surface, and this data point is just matched to the profile points in the troposphere. Visibility flag indicates that it should not be used. The vertical grid is 0, 4, 5, 6, … 44, 46, 48, …
The number of levels is 57 as indicated in the revised version of the manuscript (lines 242-243).

L260: Another approach you could have considered is averaging ln(H2O) and ln(HDO) (assuming the values are always constrained to be positive). Since there is a strong variation with height in the tropopause this avoids biasing towards large values in the average. This may explain some of the behaviour of the MIPAS-ESA HDO profile at low altitude in Fig 3.

As mentioned in lines 233-234, "the present quality assessment of H2O, HDO and $\delta D$ data mainly focuses on the stratosphere, although data for the upper troposphere and lower mesosphere are used if available".

L326: "along the stratosphere" - what does this mean? Along usually indicates a horizontal direction.

Changed to "through the stratosphere".

Fig 3: With >1000 profiles compared over most of the altitude range I think we can assume that the standard error will be negligible, so the error bars just clutter the plot.

As the standard error is very small or almost negligible, the error bars don't seem to clutter plot but make explicit the small standard errors.

Table 1: this would be clearer if the columns were lined up, eg split each into two columns, min and max, and use + signs for positive values. Also I don't think more than 1 significant figure is justified, certainly not 4 as used for the absolute bias of delta D.

The ranges in the table are expressed as the journal recommendations.
Significant figures for the absolute bias of delta-D have been modified.

L406-415: Table 1 already summarises the previous plots so I don't think yet more text summarising Table 1 is required.

We concur with the reviewer and more text summarising table 1 (table 2 in the revised version of the manuscript) is not necessary. Therefore, we have deleted this text.

Fig 5: "during I boreal"

Thank you. Corrected.

Fig 5: "The climatology is based ..." - presumably you are referring to these plots as "the climatology" but the plots are introduced as "latitude-altitude cross-sections" not as a

"climatology". Perhaps if you write "This climatology is based..." it establishes what you meant.

Thank you, it is modified.

Fig 6: These plots might be clearer if presented as deviations from the mean profile. It's hard to distinguish the various shades of blue/green which contain the signal for H2O and HDO.

We also tested the Fig 6 in terms of the deviations from the mean profile. However, we concluded that although the $H_2O$ and HDO signal could be more distinguishable, the comparison between the 3 datasets, which is the main objective of this work, weren't (see figure 4 below).

[Figure]

**Figure 4.** De-seasonalized annual cycle for (a) ACE-FTS, (b) MIPAS-IMK and (c) MIPAS-ESA datasets.

Typographcial inconsistencies

- Both upper and lower case for version, eg v8 in title, V8 in abstract

Thank you. In the revised version of the manuscript "V8" is used.

- Water vapor (eg L15) and water vapour (eg L16)

Thank you. As the journal is European, we use "vapour" in the revised version of the manuscript.

- Data set (eg title) and dataset (eg L69)

Thank you. Now "dataset" is always used.

- Use '--' in LaTeX to indicate a range of numbers, not hyphens (eg L157).

Thank you. Modified.